



# Water Cycle Acceleration in Czechia: A Water Budget Approach

Mijael Rodrigo Vargas Godoy[1], Yannis Markonis[1], Oldrich Rakovec[2, 1], Michal Jenicek[3], Riya Dutta[1], Rajani Kumar Pradhan[1], Zuzana Bešťáková[1, 4], Jan Kyselý[1, 4], Roman Juras[1], Simon Michael Papalexiou[5, 6, 1], and Martin Hanel[1]

[1]Faculty of Environmental Sciences, Czech University of Life Sciences Prague, Czechia
[2]UFZ-Helmholtz Centre for Environmental Research, Germany
[3]Department of Physical Geography and Geoecology, Charles University, Czechia
[4]Institute of Atmospheric Physics, Czech Academy of Sciences, Czechia
[5]Department of Civil Engineering, University of Calgary, Canada
[6]Global Institute for Water Security, University of Saskatchewan, Canada

**Correspondence:** M. R. Vargas Godoy (vargas_godoy@fzp.czu.cz)

**Abstract.** The water cycle in Czechia has been observed to be changing in recent years, with precipitation and evapotranspiration rates exhibiting a trend of acceleration. However, the spatial patterns of such changes remain poorly understood due to the heterogeneous network of ground observations. This study relied on multiple state-of-the-art reanalyses and hydrological modeling. Herein we propose a novel method for benchmarking hydroclimatic data fusion based on water cycle budget closure. We ranked water cycle budget closure of 96 different combinations for precipitation, evapotranspiration, and runoff using CRU TS v4.06, E-OBS, ERA5-Land, mHM, NCEP/NCAR R1, PREC/L, and TerraClimate. Then we used the best-ranked data to describe changes in the water cycle in Czechia over the last 60 years. We determined that Czechia is undergoing water cycle acceleration, evinced by increased atmospheric water fluxes. However, the increase in annual total precipitation is not as pronounced nor consistent as evapotranspiration, resulting in an overall decrease in the runoff. Furthermore, non-parametric bootstrapping revealed that only evapotranspiration changes are statistically significant at the annual scale. At higher frequencies, we identified significant spatial heterogeneity when assessing the water cycle budget at a seasonal scale. Interestingly, the most significant temporal changes in Czechia take place during spring, while median spatial patterns stem from summer changes in the water cycle.

## 1 Introduction

During the last decades, there have been significant advances in analyzing the water cycle and its response to global warming. While we expect alterations in the water cycle to respond to climate change and global warming, the actual extent and characteristics of this reaction are poorly understood (Zaitchik et al., 2023). On the one hand, small changes in total precipitation suggest a shift in precipitation towards more intense and less frequent events (Trenberth, 2011). On the other hand, it was hypothesized that an increased vertical gradient of atmospheric water vapor would offset atmospheric wind convergence in the tropics making wet regions wetter and dry regions drier (Held and Soden, 2006). Nevertheless, such claims lack conclusive



support of observed measurements and have lit the fire of controversy in the field (Vecchi et al., 2006; Allan, 2012; Skliris et al., 2016).

Undoubtedly, the advances in remote sensing observations and process-based modeling have shaped current research the most. However, as the data sources increased, it soon became apparent that large discrepancies between the data sets still

exist due to biases and uncertainties (Vargas Godoy et al., 2021). Observational data is hampered by short and heterogeneous ground-based records (Schneider et al., 2017), and unquantified uncertainties on satellite-based products (Sheffield et al., 2009). Therefore, reanalysis data providing global coverage through models while assimilating observation-based data has attained an essential role in assessing water cycle changes (Lorenz and Kunstmann, 2012). Each data source has limitations and uncertainties; when multiple sources are combined, these can compound and result in conflicting or unclear results. Hence, in

addition to uncertainty due to the complex water cycle system, which involves multiple feedback mechanisms and interactions between different components, we must account for data merge uncertainty. Accordingly, various methodologies for multi-source data integration have emerged. Among the most widely used ones are: Bayesian model averaging, constrained linear regression, neural networks, optimal interpolation, and simple weighting (Rodgers, 2000; Aires, 2014; Moazamnia et al., 2019; Pellet et al., 2019; Xiao et al., 2020). Subsequently, once merged data is generated, it is subject to post-processing for water

cycle budget closure via Monte Carlo applications and Kalman filter variations (Pan and Wood, 2006).

Several studies have quantified the water cycle by implementing data integration methods and budget closure constraints, e.g.,: Sahoo et al. (2011) integrated 16 data sets over 10 globally distributed river basins (eight for precipitation, six for evapotranspiration, one for runoff, and one for total water storage); Pan et al. (2012) integrated eight data sets over 32 globally distributed river basins (four for precipitation, two for evapotranspiration, one for runoff, and one for total water storage);

Rodell et al. (2015), integrated six data sets over continents and ocean basins (one for precipitation, three for evapotranspiration, one for runoff, and one for total water storage); Zhang et al. (2016), integrated 14 data sets globally (five for precipitation, six for evapotranspiration, one for runoff, and two for total water storage); Munier and Aires (2018) integrated 12 data sets at the global scale (four for precipitation, three for evapotranspiration, one for runoff, and four for total water storage).

The studies mentioned above focus on merging multiple data sets to end up with a single data set per water cycle component

at different spatial scales. It is evident that unconstrained uncertainty remains despite the plethora of data products derived from satellites, ground-based measurements, and climate models. This is true even for localized studies at regional scales where "ground-truth" measurements for one or more components of the water cycle are available. One region of particular interest is Czechia, a small country in Central Europe with diverse landscapes and a growing population (United Nations, 2022). The water cycle over Czechia has been experiencing significant changes in recent times, affecting various aspects of

the water balance in the region, including changes in river flow regimes and water quality, loss of wetlands, and changes in the frequency and severity of extreme events (Mozny et al., 2020). Besides, changes in the rainfall-snowfall partition have given rise to a decrease in snow cover and premature snowmelt (Nedelcev and Jenicek, 2021). These changes in the water cycle are expected to continue in the near-future (Kyselý and Beranová, 2009; Jenicek et al., 2021). Precipitation, in particular, is expected to increase its mean mainly in winter and extreme rates throughout the year (Kyselý et al., 2011). In addition,

increased human activities, such as urbanization and agriculture, have led to changes in land use and land cover, which in turn





has contributed to the occurrence of floods and droughts (Svoboda et al., 2016). Droughts, have had disastrous consequences for agriculture, forestry, water management, and other human activities (Brázdil et al., 2009). Consequently, the water cycle in Czechia and human activity find themselves on a causal feedback loop.

In this study, we aim to estimate the water cycle changes over Czechia between the 1961-1990 and 1991-2020 periods, and determine the current trends and patterns in water cycle components. Our analysis includes various data sets at different spatiotemporal scales allowing us to assess 96 data combinations for budget closure. Rather than enforcing budget closure on a multi-source integrated data set or assessing different integration methods, we explored an empirical method to rank how multiple data set combinations close the water cycle budget while correlating to referential data estimates of individual water cycle components. In this manner, we are not generating yet another new data set but are identifying the best combination among the data sets available for our study domain. Only the data sets with the best rankings as determined by our proposed benchmarking were used in all subsequent computations. We found that hydroclimatic models, as expected, have better water budget closure. However, ERA5-Land is not far off despite known non-closure limitations associated with reanalyses. We identified an overall acceleration of atmospheric water fluxes. Simultaneously, we report a heterogeneous distribution of freshwater availability.

## 2 Data and Methods

### 2.1 Study Area

Czechia is a landlocked (surrounded by Germany, Austria, Slovakia, and Poland) European country that covers an area of 78 864 km$^2$. Czechia is an essential headwaters region of the European continent; in hydrological terms, it can be called the roof of Europe. The country is home to several large rivers, including the Vltava, the Elbe, the Morava, and the Oder, all of which have their sources within it. Czechia is situated at the intersection of three sea drainage basins: the North Sea, the Baltic Sea, and the Black Sea, which, in return, divide Czechia into three main hydrological catchment areas: the Elbe, Oder, and Danube basins (Figure 1). All of these major watercourses drain water into neighboring states. The water sources of Czechia are thus almost exclusively dependent on precipitation.

### 2.2 Data

To assess water cycle acceleration we gathered data sets with at least 60 years of record. This first filter reduced the plethora of publicly available data sets to nine data sets from multiple sources (observation-based, reanalysis, and hydrological model products) plus three evaluation references (Table 1). The evaluation data sets for precipitation, evapotranspiration, and runoff are the Czech Hydrometeorological Institute (CHMI), Global Land Evaporation Amsterdam Model (GLEAM v3.6a; Martens et al. (2017)), and GRUN (Ghiggi et al., 2019), respectively. Six precipitation data sets: Climatic Research Unit at the University of East Anglia (CRU TS v4.06; Harris et al. (2020)), European Centre for Medium-Range Weather Forecasts (ECMWF) Reanalysis (ERA5-Land; Muñoz-Sabater et al. (2021)), the E-OBS data set from the Copernicus Climate Change Service (Cornes et al., 2018), National Centers for Environmental Prediction & the National Center for Atmospheric Research Re-





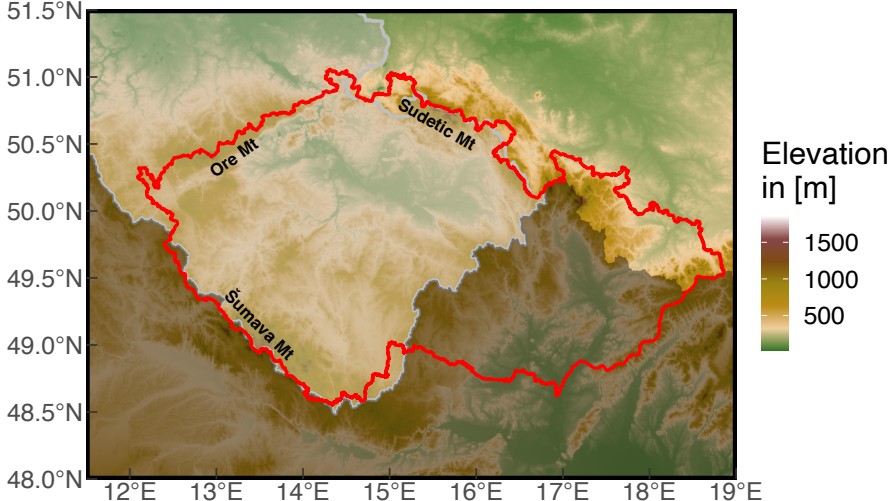

**Figure 1.** The three drainage basins within Czechia's boundaries. Elbe (light gray shade), Danube (dark gray shade), and Oder (no shade).

analysis One (NCEP/NCAR R1; Kalnay et al. (1996)), Precipitation Reconstruction Over Land (PREC/L; Chen et al. (2002)), and TerraClimate (Abatzoglou et al., 2018). Note that, E-OBS (hereinafter mHM(E-OBS)) was used as meteorologic input for the mesoscale Hydrologic Model (mHM; Samaniego et al. (2010); Kumar et al. (2013))). Four evapotranspiration data sets:
ERA5-Land, mHM, NCEP/NCAR R1, and TerraClimate. Four runoff data sets: ERA5-Land, mHM, NCEP/NCAR R1, and TerraClimate. Using the above listed data sets we assessed a total of 96 different combinations.

**Table 1.** Data set description. $P$ is precipitation, $E$ is evapotranspiration, and $Q$ is runoff.

| Name | Variable(s) | Spatial Resolution | Temporal Resolution | Record Length | Reference |
|------|-------------|--------------------|--------------------|---------------|-----------|
| CHMI | $P$ | Point | Daily | 1961-2020 | http://portal.chmi.cz |
| CRU TS v4.06 | $P$ | 1° | Monthly | 1901-2020 | Harris et al. (2020) |
| E-OBS | $P$ | 0.125° | Daily | 1950-2020 | Cornes et al. (2018) |
| ERA5-Land | $P, E, Q$ | 0.1° | Monthly | 1950-2020 | Muñoz-Sabater et al. (2021) |
| GLEAM v3.6a | $E$ | 0.25° | Daily | 1980-2020 | Martens et al. (2017) |
| GRUN | $Q$ | 0.5° | Monthly | 1902-2014 | Ghiggi et al. (2019) |
| mHM | $E, Q$ | 0.125° | Daily | 1950-2020 | Samaniego et al. (2010) |
| NCEP/NCAR R1 | $P, E, Q$ | T62 | Monthly | 1948-2020 | Kalnay et al. (1996) |
| PREC/L | $P$ | 0.5° | Monthly | 1948-2020 | Chen et al. (2002) |
| TerraClimate | $P, E, Q$ | 4 km | Monthly | 1958-2020 | Abatzoglou et al. (2018) |





### 2.2.1 Evaluation References

The Czech Hydrometeorological Institute (CHMI) provides station derived precipitation data. The CHMI station network consists of approximately 700 stations distributed with a mean density of one station per each 100 km$^2$, adequately representing the distinct geographical features of Czechia (Kašpar et al., 2021). Although the data collection and related services for a specific station are generally managed by the regional branches of CHMI, the entire territory station data can be accessed from the Department of Climatology of CHMI at once. All the data sets are undergone robust quality control checks by CHMI before being added to the database. Herein, we gathered the country level estimates calculated by CHMI (one value per month) for a period of 60 years (1961-2020).

The Global Land Evaporation Amsterdam Model (GLEAM) is a satellite-based global evaporative model designed to estimate terrestrial evapotranspiration from 1980 to the near present (Martens et al., 2017). It encompasses a set of algorithms that estimates a total of 11 variables, including actual and potential evapotranspiration rates. Briefly, the estimation procedure consists of two major steps. First, given the temperature and radiation data sets, potential evapotranspiration rates in [mm/day] are estimated by land cover type using the Priestley and Taylor equation (Priestley and Taylor, 1972). Second, potential evapotranspiration is converted into actual evapotranspiration based on an evaporative stress factor. Despite GLEAM not being a ground station derived product, or even a fully observation-based data set, previous studies have extensively evaluated GLEAM and advocate for its high quality (Yang et al., 2017; Bai and Liu, 2018; Liu et al., 2021).

GRUN is a gridded global monthly runoff reconstruction generated by a machine learning model (Ghiggi et al., 2019). It is available at a 0.5° spatial resolution and monthly time step from 1902 to 2014. Conceptually, the machine learning algorithm was trained with observations of monthly temperature, precipitation and streamflow. Then, the trained model is used to predict monthly runoff at ungauged catchments. Although the method does not implement any physically detailed hydrological model, the predicted runoff showed better results than the ensemble mean of 13 global hydrological model simulations when compared to observational references. Furthermore, GRUN has been extensively applied in regions with sparse in-situ measurements (Hu et al., 2021; Xiong et al., 2022; Xu et al., 2022; Mei et al., 2023).

### 2.2.2 Observational-based Products

CRU TS is a popularly used gridded data set generated by the University of East Anglia's Climate Research Unit (Harris et al., 2020). It is known for its historical long-term coverage, which is available from 1901 to the near present. The data set comes with a 0.5° spatial resolution at the monthly scale. It compiles station data from multiple sources such as the Food and Agricultural Organisation (FAO), the World Meteorological Organisation (WMO), and the National Meteorological Agencies (NMA's) (Sun et al., 2018). CRU TS v4, its latest version, implemented angular distance based interpolation to facilitate tracing back the stations upon which the gridded data set has been constructed.

PREC/L, created by the US Climate Prediction Center (CPC), is a gridded product entirely based on the station data set (Chen et al., 2002) with global coverage and monthly time step. PREC/L draws data from over 17 000 stations from the Global Historical Climatology Network version2 (GHCN v2; Peterson and Vose, 1997) and the Climate Anomaly Monitoring





System (CAMS; Janowiak and Xie, 1999). Subsequently, the data is interpolated to construct the gridded product at three different resolutions (0.5°, 1°, and 2.5°). Herein, we used the 0.5° monthly precipitation, whose record extends from 1948 to the present.

### 2.2.3 Hydrological Models

The mesoscale Hydrologic Model (mHM; Samaniego et al., 2010; Kumar et al., 2013) is a conceptual grid-based model rep-
resenting dominant hydrological fluxes and storage at the Earth's surface and subsurface through a set of ordinary differential equations. mHM represents processes such as interception, snow, soil moisture, evapotranspiration, and various runoff components like fast/slow interflow and baseflow. The model was established, parameterized and evaluated over the European continent (Rakovec et al., 2016b; Samaniego et al., 2019; Rakovec et al., 2022). The meteorological inputs were based on daily E-OBS data (Cornes et al., 2018) of precipitation in addition to minimum, maximum and average temperature. The potential
evapotranspiration was derived using the method of (Hargreaves and Samani, 1982). The spatial resolution of the model grid corresponds to 0.125°.

Terraclimate is a high-resolution gridded global climate data set that provides the mean climate and mean water balance data covering a time span of 1958 to the present (Abatzoglou et al., 2018). The data set is commonly known for its high spatial resolution (4 km). It uses various global gridded climate data sets such as WorldClim v2 (Fick and Hijmans, 2017) and v1.4
(Hijmans et al., 2005), CRU TS v4 (Harris et al., 2020), Japanese 55-year Reanalysis (JRA55) (Kobayashi et al., 2015), and Root zone storage capacity (Wang-Erlandsson et al., 2016) in order to generate the high-resolution monthly climate variables time series at the global level. An additional advantage of the Terraclimate is that it produces monthly surface water balance based on a water balance model along with primary climatic variables such as temperature, precipitation, solar radiation, etc.

### 2.2.4 Reanalyses

ERA5-Land is the latest fifth-generation global atmospheric reanalysis product developed by the European Center for Medium-Range Weather Forecast (ECMWF) (Muñoz-Sabater et al., 2021). ERA5-Land, as the name implies, builds upon the terrestrial component of ERA5 and downscales the model spatial grid resolution from 31 km into 9 km. As a result, ERA5-Land delivers either hourly or monthly estimates with a spatial resolution of 0.1°. Given its high spatiotemporal resolution and long record, ERA5-Land provides valuable data for comprehensive analysis and diverse hydrological applications at the global scale.

The NCEP/NCAR Reanalysis project one is produced by the collaboration between the National Centers for Environmental Prediction (NCEP) and the National Center for Atmospheric Research (NCAR) (Kalnay et al., 1996). It is the longest-running reanalysis that uses rawindsonde data, at the expense that the model and data assimilation scheme are antiquated Trenberth et al. (2011). The data set is distributed on a T62 Gaussian grid (approximately 1.875° at the equator) and its record start dates back to 1948.





## 2.3 Data Evaluation

We validated the gathered data sets to capture the temporal variability of water cycle components as described by the three observational references via:

- The coefficient of determination (R-squared or $R^2$)

$$R^2 = 1 - \frac{\sum_i^n (y_i - \hat{y}_i)^2}{\sum_i^n (y_i - \overline{y})^2}$$

where $i$ starts on the first year of the available record, $n$ is the last year of the available record, $y_i$ is the observational reference on year $i$, $\hat{y}_i$ is the data set estimate on year $i$, and $\overline{y}$ is the mean observational estimate for the full available
record.

- Root Mean Square Error (RMSE)

$$\text{RMSE} = \sqrt{\frac{\sum_i^n (y_i - \hat{y}_i)^2}{N}}$$

where $i$ starts on the first year of the available record, $n$ is the last year of the available record, $y_i$ is the observational reference on year $i$, $\hat{y}_i$ is the data set estimate on year $i$, and $N$ is the total number of years in the full available record.

All data sets were spatial weighted averaged over Czechia and temporally aggregated to an annual scale over the calendar year. Note that only precipitation data sets could be evaluated over the entire 60-year period of 1961-2020. In contrast, evapo-
165 transpiration was evaluated over 1980-2020 and runoff over 1961-2014. In order to compare a 30-year mean among all water cycle components, the common period of 1981-2010 was selected.

### 2.4 Data Set Ranking

A success metric widely used among several studies is getting the budget closure residual ($R$) as close to zero as possible. Herein, we define the budget closure residual as follows:

$$R = P - E - Q \tag{1}$$

where $P$ is precipitation, $E$ is evapotranspiration, and $Q$ is runoff. Thus, we have 96 distributions of 60 annual values each. The ranking of a given data set combination was determined via:

$$Ranking = \frac{|\overline{R_i}|\sigma_{R_i}}{(cor(P_i - E_i, Q_i)cor(P_i, P_\text{o})cor(E_i, E_\text{o})cor(Q_i, Q_\text{o}))^2} \tag{2}$$

where $|\overline{R_i}|$ is the absolute value of the mean of the 60 annual residuals for the $i$-th combination, $\sigma_{R_i}$ is the standard deviation of the 60 annual residuals for the $i$-th combination, $cor(P_i - E_i, Q_i)$ is the correlation between $P - E$ and $Q$ for the $i$-th combination, $cor(P_i, P_\text{o})$ is the correlation between $P$ of the $i$-th combination and the precipitation evaluation reference, $cor(E_i, E_\text{o})$ is the correlation between $E$ of the $i$-th combination and the evapotranspiration evaluation reference, and $cor(Q_i, Q_\text{o})$ is the





correlation between $Q$ of the $i$-th combination and the runoff evaluation reference. The ranking method proposed herein can easily be applied to any other referential data set for evaluation. In data-limited areas or those with a poor observational network, the ranking method may still be applied using external data as an evaluation reference, or the corresponding term in the equation can be simply left out. E.g., if evapotranspiration data for evaluation is not available, Equation 2 becomes:

$$Ranking = \frac{|\overline{R_i}|\sigma_{R_i}}{(cor(P_i - E_i, Q_i)cor(P_i, P_o)cor(Q_i, Q_o))^2}$$

In the case of Czechia, we used GLEAM v3.6a as the evaluation reference, due to the absence of access to observational evapotranspiration.

## 2.5 Water Cycle Changes

We assessed the empirical distribution of spatial weighted average values (accounting for the area of each grid cell in proportion to the total area being averaged) of annual water cycle fluxes between 1961-1990 and 1991-2020 for three of the best data set combinations. To account for the influence of extreme value in the latter period due to the 100-year drought of 2003 (Brázdil et al., 2013), we compared the median values rather than their means. To deepen our assessment of changes in the distribution of water cycle fluxes, we compared their monthly values between 1961-1990 and 1991-2020. To determine the statistical significance of the above-mentioned changes, we employed non-parametric bootstrapping of 10 000 iterations. Subsequently, we performed an analogous analysis in space. We computed the change in the median values between 1961-1990 and 1991-2020 over each grid cell. Note that each data set was assessed at its native resolution for this part of the analysis. Finally, we examined the change patterns of water cycles through the seasons. Herein, we considered: winter as December, January, and February; spring as March, April, and May; summer as June, July, and August; autumn as September, October, and November.

## 3 Results

### 3.1 Benchmarking water cycle components

Our analysis describes the most recent spatiotemporal changes on the water cycle in Czechia. For starters, we examined precipitation, evapotranspiration, and runoff estimates from the gathered data sets compared to CHMI (Figure 2a), GLEAM v3.6a (Figure 2b), and GRUN v1 (Figure 2c) as the respective evaluation references. The variability of estimates from precipitaion and runoff data sets (Figure 2a and c) visibly have a broader spread than those of evapotranspiration (Figure 2b). While one may suspect the spread in precipitation is due to the higher number of data sets available, they correlate better to their evaluation reference than evapotranspiration or runoff. The data set with the highest correlation values for precipitation is mHM(E-OBS) with R-squared of approximately 0.99 (Figure 2a). mHM has the highest correlation for runoff, with R-squared circa 0.86 (Figure 2c), falling to the second highest for evapotranspiration (R-squared 0.7; Figure 2c). Interestingly, the values for the 30-year average in mHM underestimates runoff (Figure 2c) but overestimates evapotranspiration (Figure 2b). In contrast, NCEP/NCAR R1 consistently reports the lowest correlation values regardless of the water flux of interest. Additionally, other than for runoff,





it has considerably higher RMSE values than the rest of the data sets. To some degree, ERA5-Land is the in-betweener data
set because it has high correlation values and simultaneously has high RMSE for precipitation and evapotranspiration, yet for
runoff, ERA5-Land exhibits moderate correlation and small RMSE.

It would be sensible to use the best data set for each water flux to proceed with further analysis. However, we first verified
if the best data sets individually would depict the best water cycle budget in conjunction. Conventional metrics like R-squared
and RMSE cannot be directly applied to a combination of data sets. We defined an empirical ranking metric, as described by
Equation 2, where the smallest the value, the better the data set combination. While our ranking approach is empirical and
simple, Equation 2 correctly identifies narrow distribution centered mean zero with higher ranked positions compared to wider
distributions centered around positive or negative values (Figure 3). Upon ranking all 96 possible combinations (Table 2), we
observe that even though mHM outperformed TerraClimate for individual water flux estimates, the TerraClimate exclusive
combination offers the best water budget closure. We expected combinations with hydrological model data to be highly ranked
and reanalyses to be poorly ranked due to the above-reported considerable biases of the latter. Notwithstanding, we were
surprised to see the ERA5-Land exclusive combination (i.e., all flux estimates from the same data set) among the top five ranks.
The first combination that includes at least one estimate from NCEP/NCAR R1 is at the 45th rank, and the NCEP/NCAR R1
exclusive combination is at the 90th rank.

### 3.2 Temporal changes in the water cycle

Moving forward, we computed the change in water fluxes' annual distribution via shifts on their 30-year median (Figure 4).
Also, we assessed the statistical significance of the observed change in the medians by non-parametric bootstrapping (10 000
iterations). Hereupon, we will report results only for the first- (TerraClimate exclusive), second- (mHM exclusive), and fifth-
ranked (ERA5-Land exclusive) data combinations. Because the third- (CRU TS v4.06, TerraClimate, Terraclimate) and fourth-
ranked (mHM, mHM, TerraClimate) data combinations have a single data set different from the first- and second-ranked ones,
as such, we would be showing the same plots and statistics multiple times. TerraClimate and mHM show similar increases in
precipitation and evapotranspiration circa 20 mm, but only evapotranspiration manifests a statistically significant change (p
< 0.01). Evapotranspiration changes underwhelming those of precipitation stand further accentuated in ERA5-Land, whose
magnitude of the change in evapotranspiration is almost 60 mm and in precipitation is less than -1 mm. Another peculiarity of
ERA5-Land is that runoff, with a change of -56 mm at p = 0.01 statistical significance. Regarding the estimates for precipitation
minus evapotranspiration, we observe three different behaviors: TerraClimate has a change in P-E in the opposite direction of
runoff (1 mm vs. -5 mm); mHM has a change in P-E of smaller magnitude than runoff (-2 mm vs. -9 mm); ERA5-Land has
similar changes for both P-E and runoff (-55 mm vs. -56 mm), but with values one order of magnitude higher than those of
TerraClimate and mHM.

The above results, seemingly disagreeing with the expected increases reported in previous literature (Kyselý and Beranová,
2009; Svoboda et al., 2016; Kašpárek and Kožín, 2022), indicate that there have not been any statistically significant changes
in median annual precipitation over Czechia between the last two 30-year periods. Thereafter, we proceeded to look into
changes between 1961-1990 and 1991-2020 monthly water fluxes (Figure 5). Note that hereinafter we mention only months





**Figure 2.** Benchmarking spatial weighted average annual water fluxes over Czechia between 1961 and 2020. For consistency and comparability between different water fluxes, annual anomalies were computed using the 1981-2010 average as a reference, the common period among all data sets. The 1981-2010 average and standard deviation are listed at the bottom left of each panel. Linear correlation summary statistics are displayed at the bottom right of each panel. The spread of the estimates being evaluated is shown in gray, and their mean is in white. (a) Precipitation evaluation. CHMI data is shown in blue. (b) Evapotranspiration evaluation. GLEAM v3.6a is shown in green. (c) Runoff evaluation. GRUN v1 data is shown in purple.





**Table 2.** Data set ranking as determined by Equation 2. $P$ is precipitation, $E$ is evapotranspiration, $Q$ is runoff, $\bar{R}$ is the mean residual over 60 years, $\sigma_R$ is the standard deviation of the residual over 60 years, $cor(P-E, Q)$ is the correlation between $P-E$ and $Q$ for the $i$-th ranked combination, $cor(P, P_o)$ is the correlation between $P$ of the $i$-th ranked combination and CHMI, $cor(E, E_o)$ is the correlation between $E$ of the $i$-th ranked combination and GLEAM v3.6a, and $cor(Q, Q_o)$ is the correlation between $Q$ of the $i$-th ranked combination and GRUN v1.

| Ranking | $P$ | $E$ | $Q$ | $\bar{R}$ | $\sigma_R$ | $cor(P-E, Q)$ | $cor(P, P_o)$ | $cor(E, E_o)$ | $cor(Q, Q_o)$ |
|---|---|---|---|---|---|---|---|---|---|
| 1st | TerraClimate | TerraClimate | TerraClimate | -0.346 | 30.204 | 0.846 | 0.941 | 0.532 | 0.820 |
| 2nd | mHM(E-OBS) | mHM | mHM | -0.912 | 51.231 | 0.816 | 0.994 | 0.835 | 0.926 |
| 3rd | CRU TS v4.06 | TerraClimate | TerraClimate | -1.749 | 29.944 | 0.843 | 0.938 | 0.532 | 0.820 |
| 4th | mHM(E-OBS) | mHM | TerraClimate | 7.604 | 56.406 | 0.754 | 0.994 | 0.835 | 0.820 |
| 5th | ERA5-Land | ERA5-Land | ERA5-Land | -5.554 | 66.606 | 0.701 | 0.951 | 0.887 | 0.786 |
| 6th | PRECL/L | mHM | mHM | 8.498 | 61.887 | 0.633 | 0.891 | 0.835 | 0.926 |
| ... | ... | ... | ... | ... | ... | ... | ... | ... | ... |
| 11th | PRECL/L | mHM | TerraClimate | 17.013 | 60.281 | 0.658 | 0.891 | 0.835 | 0.820 |
| ... | ... | ... | ... | ... | ... | ... | ... | ... | ... |
| 24th | ERA5-Land | ERA5-Land | TerraClimate | 64.008 | 65.157 | 0.748 | 0.951 | 0.887 | 0.82 |
| ... | ... | ... | ... | ... | ... | ... | ... | ... | ... |
| 45th | NCEP/NCAR R1 | ERA5-Land | ERA5-Land | 2.3 | 193.986 | 0.429 | 0.181 | 0.887 | 0.786 |
| ... | ... | ... | ... | ... | ... | ... | ... | ... | ... |
| 48th | PRECL/L | TerraClimate | ERA5-Land | -91.688 | 61.887 | 0.416 | 0.891 | 0.532 | 0.786 |
| ... | ... | ... | ... | ... | ... | ... | ... | ... | ... |
| 72nd | mHM(E-OBS) | NCEP/NCAR R1 | TerraClimate | -312.910 | 59.149 | 0.723 | 0.994 | 0.073 | 0.820 |
| ... | ... | ... | ... | ... | ... | ... | ... | ... | ... |
| 90th | NCEP/NCAR R1 | NCEP/NCAR R1 | NCEP/NCAR R1 | 292.024 | 137.297 | 0.675 | 0.181 | 0.073 | 0.465 |
| ... | ... | ... | ... | ... | ... | ... | ... | ... | ... |
| 96th | CRU TS v4.06 | NCEP/NCAR R1 | NCEP/NCAR R1 | -424.772 | 93.962 | -0.019 | 0.938 | 0.073 | 0.465 |

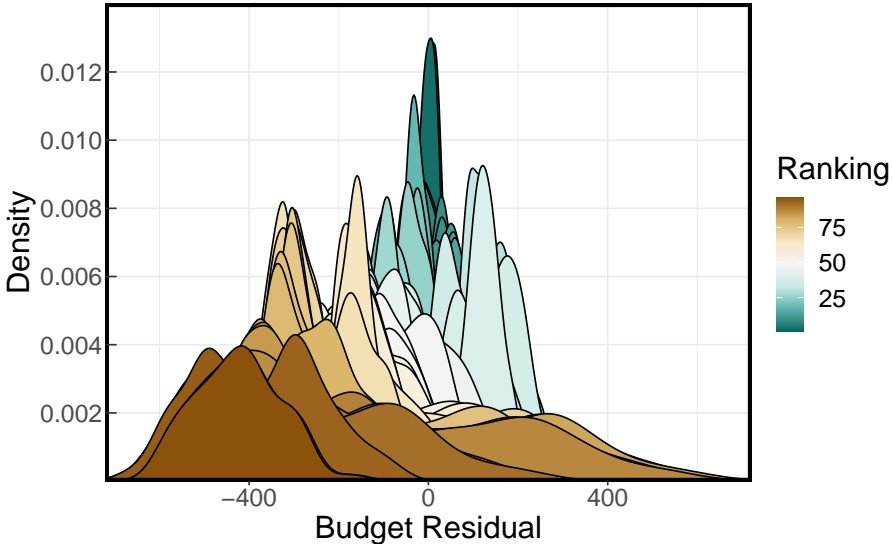

**Figure 3.** Empirical distribution of all possible 96 data set combinations colored based on their ranking as determined by Equation 2. The color gradient goes from higher ranked combinations colored in shades green to lower ranked combinations colored in shades of brown.

with statistically significant changes (p < 0.01). Regarding precipitation, we observe a consistent increase of around 14 mm during October and circa 11 mm during July present in TerraClimate, mHM(E-OBS), and ERA5-Land. Besides, mHM(E-OBS)
and ERA5-Land had decreasing changes in April of -6 mm and -9 mm, respectively. We also found a -5 mm decrease during November, present only in mHM(E-OBS). In terms of evapotranspiration, as expected from the statistically significant changes described for annual values, we report increases between 1-10 mm depending on the month. TerraClimate has the shortest period of continuous changes with gradually increasing magnitude from January (1 mm) to March(9 mm). mHM on top of said evapotranspiration behavior from January (1 mm) to April (4 mm) also shows the subsequent oscillating behavior: May
(2 mm), June (2 mm), July(4 mm), and August (3 mm). ERA5-Land changes in evapotranspiration have a behavior similar to mHM but with overall higher magnitudes and two months longer. I.e., a consecutive increase from December (1 mm) to April (8 mm) and subsequent swings back and forth: May (7 mm), June (7 mm), July(10 mm), August (8 mm), and September (3 mm). Concerning runoff, there is a striking unique visual for TerraClimate, whose range of values from February to Abril is considerably larger than those of mHM or ERA5-Land. A runoff decrease is present in all data sets for April and May, with an
added magnitude of -18 mm, -8 mm, and -12 mm for TerraClimate, mHM, and ERA5-Land, respectively. Interestingly, these runoff decreases are translated only into mHM and ERA5-Land through precipitation minus evapotranspiration decrease in April (-6 mm and -15 mm).

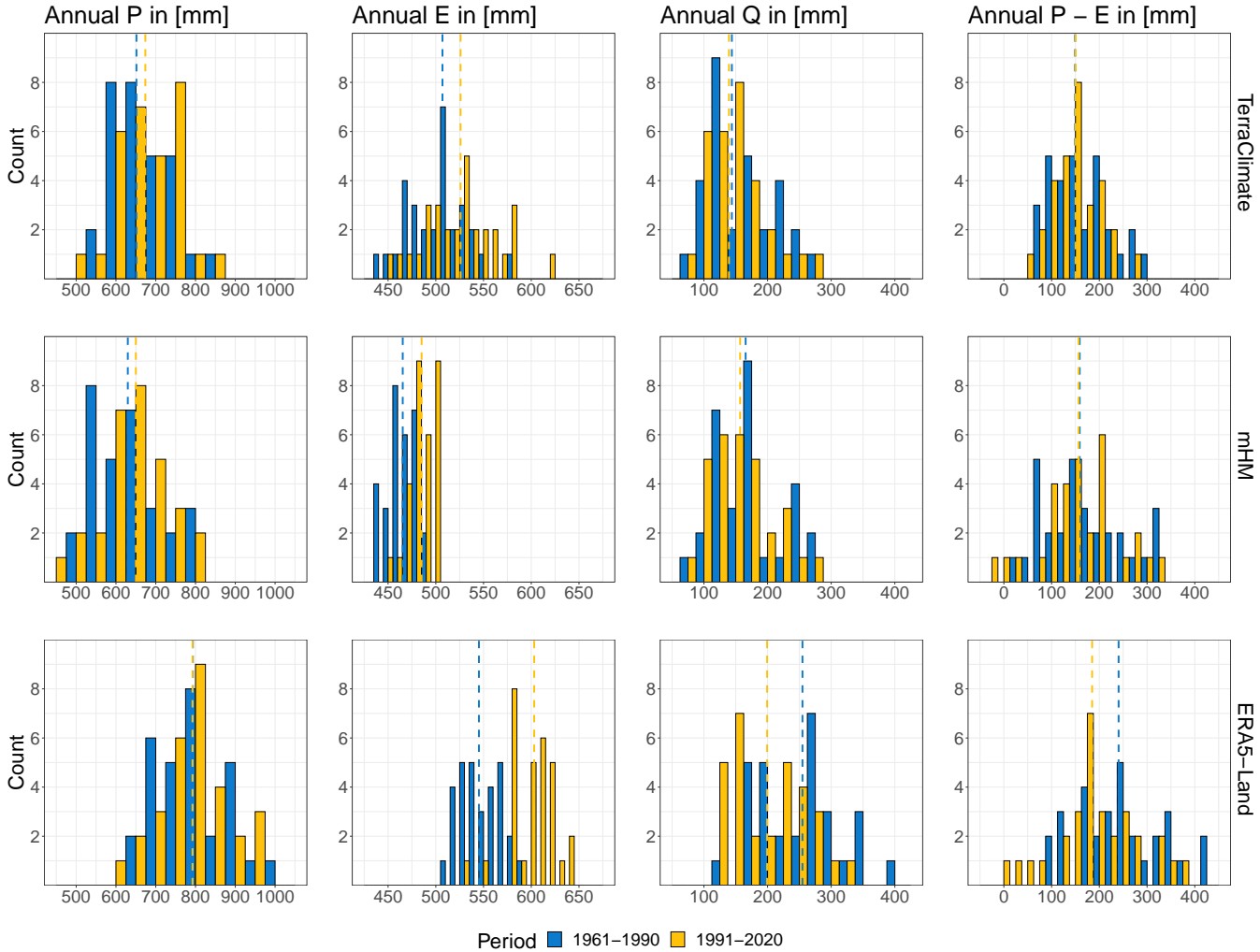

**Figure 4.** Histograms of spatial weighted average annual water fluxes over Czechia, where $P$ is precipitation, $E$ is evapotranspiration, $Q$ is runoff, and $P - E$ is precipitation minus evapotranspiration. Data are divided into two 30-year periods: 1961-1990 (light gray) and 1991-2020 (dark gray). The median value of each 30-year period is represented by dashed lines in their respective color. Top row: TerraClimate ($P$), TerraClimate ($E$), and TerraClimate ($Q$). Middle row: mHM(E-OBS) ($P$), mHM ($E$), and mHM ($Q$). Bottom row: ERA5-Land ($P$), ERA5-Land ($E$), and ERA5-Land ($Q$).



**Figure 5.** Box plot of spatial weighted average monthly water fluxes over Czechia, where $P$ is precipitation, $E$ is evapotranspiration, $Q$ is runoff, and $P - E$ is precipitation minus evapotranspiration. Data are divided into two 30-year periods: 1961-1990 (light gray) and 1991-2020 (dark gray). Left column: TerraClimate ($P$), TerraClimate ($E$), and TerraClimate ($Q$). Middle column: mHM(E-OBS) ($P$), mHM ($E$), and mHM ($Q$). Right column: ERA5-Land ($P$), ERA5-Land ($E$), and ERA5-Land ($Q$).





### 3.3 Spatial patterns of water cycle changes

The results shown so far provide insight into the temporal changes water cycle components have undergone in the past 60 years, considering spatial weighted averaged values across Czechia. To expand our analysis from the temporal into the spatial domain and provide insight into the spatiotemporal features of the selected data sets, we mapped the difference between the 1991-2020 and the 1961-1990 medians for $P$, $E$, $Q$, and $P - E$ (Figure 6). Note that maps for each product were generated at their native resolutions, i.e., TerraClimate at 4 km, mHM at $0.125°$, and ERA5-Land at $0.1°$. At first glance, we observe overall agreement in spatial patterns between data sets for evapotranspiration and runoff, with slight discrepancies around the Sudetic (northeast), Šumava (southwest), and Ore (northwest) Mountains. In particular, ERA5-Land exhibits changes of higher magnitude in evapotranspiration (increase) and runoff (decrease) than TerraClimate and mHM.

Contrary to the above-described agreement, there is no consensus on spatial precipitation patterns among data sets. We discern three different patterns: TerraClimate shows a homogeneous increase across the country with a particular contour of higher increase that starts at the Šumava Mountains and diminishes toward the Ore Mountains and a slight decrease around the Sudetes; ERA5-Land portrays a somewhat zonal pattern with increasing bands north of $50.5°$N and south of $49.5°$N of the country and a decreasing band in the middle; mHM pattern is in between those of TerraClimate and ERA5-Land, with the band of precipitation decrease being smaller than that of ERA5-Land confined west of $15°$E. While some of these heterogeneities are echoed in $P - E$ spatial patterns, there is a general decrease across data sets over Czechia. Therefore, evapotranspiration changes appear to dominate the spatial distribution of water availability.

Based on the results observed in Figure 5, we have previously identified that monthly patterns of increase or decrease in water fluxes are, to some extent, aligned with their seasonal variability. Thus this time around, we aggregated the data seasonally rather than looking at the monthly spatial distribution of changes in the median between the two 30-year periods. While individual characteristics for each data set are further emphasized by looking into seasonal spatial patterns, we identify some common traits. A dominant pattern of precipitation decrease is localized to the Westernmost part of Czechia during winter and expands to the rest of the country during spring. Evapotranspiration increases of the highest magnitude take place during spring and summer. As a result of this opposing direction, during spring, we see the most substantial decrease in runoff and $P - E$ therein. Furthermore, it is safe to state that if evapotranspiration generally increases despite decreasing patches of precipitation (present to a greater or lesser extent across all seasons), the water cycle in Czechia is dominated by changes in energy rather than water availability.

TerraClimate, with a resolution of 4 km, offers far more detail on spatial patterns than other data sets (Figure 7). It has a semester split for precipitation, with a decreasing pattern dominating winter and spring and an increasing pattern dominating summer and autumn. Evapotranspiration decreases during spring and summer but does not cover nearly as much area of Czechia as precipitation when decreasing. Runoff changes circumscribe winter (increase) and spring (decrease) and are relatively mute during summer and autumn. Regarding water availability, the patterns of $P - E$ reflect those of precipitation. However, the increases in summer and autumn are not as notable. Autumn is a season of spatial homogeneity in TerraClimate because precipitation, evapotranspiration, runoff, and $P - E$ all depict countrywide increases, albeit of smaller magnitude than

**Figure 6.** Spatial pattern of changes in median water fluxes over Czechia between two 30-year periods: 1961-1990 and 1991-2020. I.e., the value of each grid cell is equal to the median value of 1991-2020 minus the median value of 1961-1990. $P$ is precipitation, $E$ is evapotranspiration, $Q$ is runoff, and $P - E$ is precipitation minus evapotranspiration. Left column: TerraClimate ($P$), TerraClimate ($E$), and TerraClimate ($Q$). Middle column: mHM(E-OBS) ($P$), mHM ($E$), and mHM ($Q$). Right column: ERA5-Land ($P$), ERA5-Land ($E$), and ERA5-Land ($Q$).





in other seasons. On the other hand, a distinctive contrast takes place in winter, in which we have a decrease in runoff in spite of an increase in water availability.

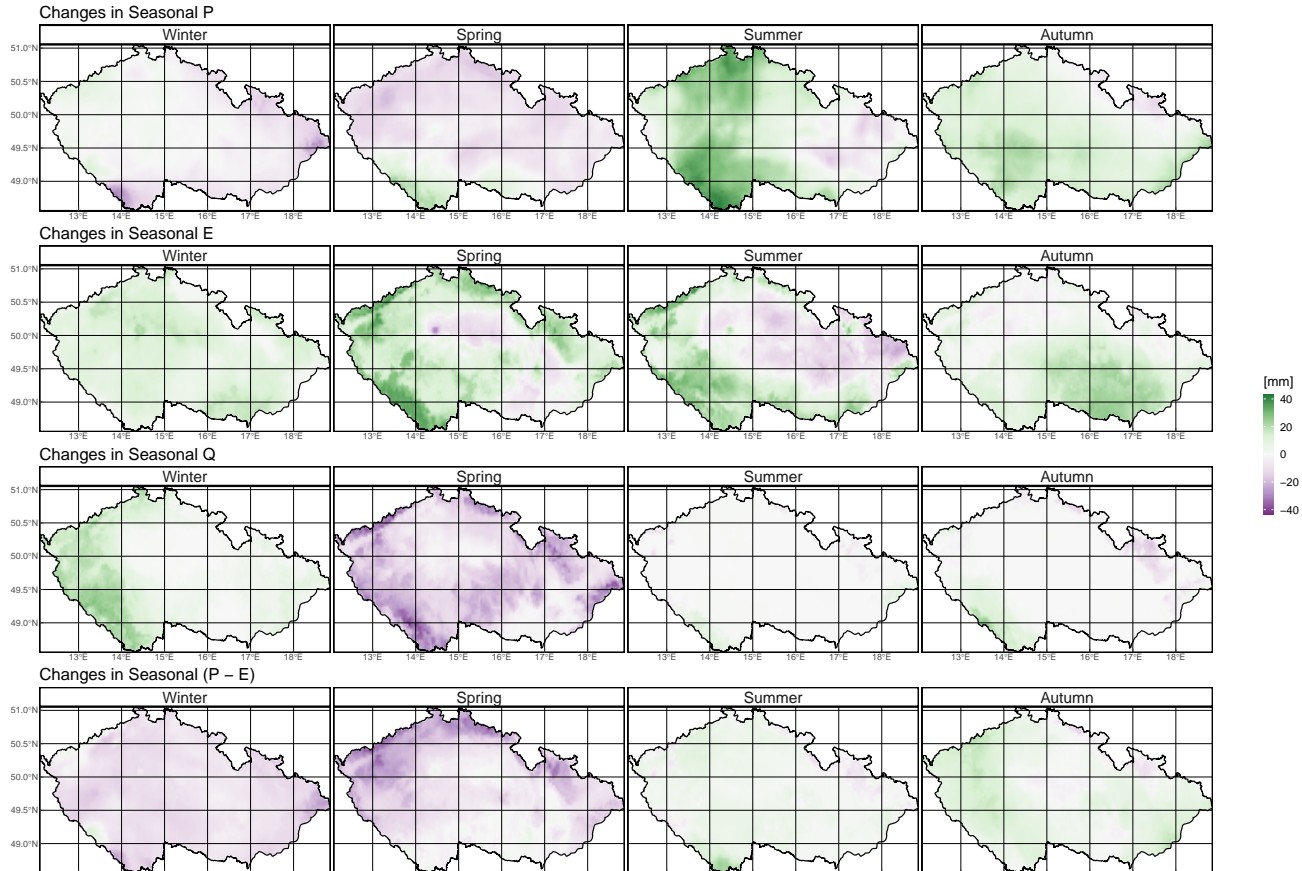

**Figure 7.** TerraClimate spatial pattern of changes in seasonal median water fluxes over Czechia between two 30-year periods: 1961-1990 and 1991-2020. I.e., the value of each grid cell is equal to the seasonal median value of 1991-2020 minus the seasonal median value of 1961-1990. $P$ is precipitation, $E$ is evapotranspiration, $Q$ is runoff, and $P - E$ is precipitation minus evapotranspiration. The seasons are defined as follows: winter as December, January, and February; spring as March, April, and May; summer as June, July, and August; autumn as September, October, and November.

Seasonal spatial patterns of mHM have the least substantial changes, with magnitudes mainly in the -25 mm to 25 mm
range compared to the -40 mm to 40 mm range of TerraClimate and ERA5-Land (Figure 8). Precipitation patterns mimic those of TerraClimate except for autumn, where mHM(E-OBS) holds more heterogeneity. Contemporaneously, we observe slightly decreased evapotranspiration. For the rest of the seasons, evapotranspiration presents a widespread pattern of positive changes, with the highest magnitudes in summer. There is a dominant decreasing pattern for runoff across all seasons. In winter, there are pinpoint increases around the Czech borders near the Sudetic, Šumava, and Ore Mountains. $P - E$ has the highest magnitude



for decreasing change in spring. There is a mixed pattern of increase and decrease for $P - E$ in winter and summer, yet the

extent of decreasing changes is more prominent. Once again, analogous to TerraClimate, we find a season of contrasting runoff

(decreasing) and $P - E$ (increasing) changes, but for mHM, it takes place in autumn.

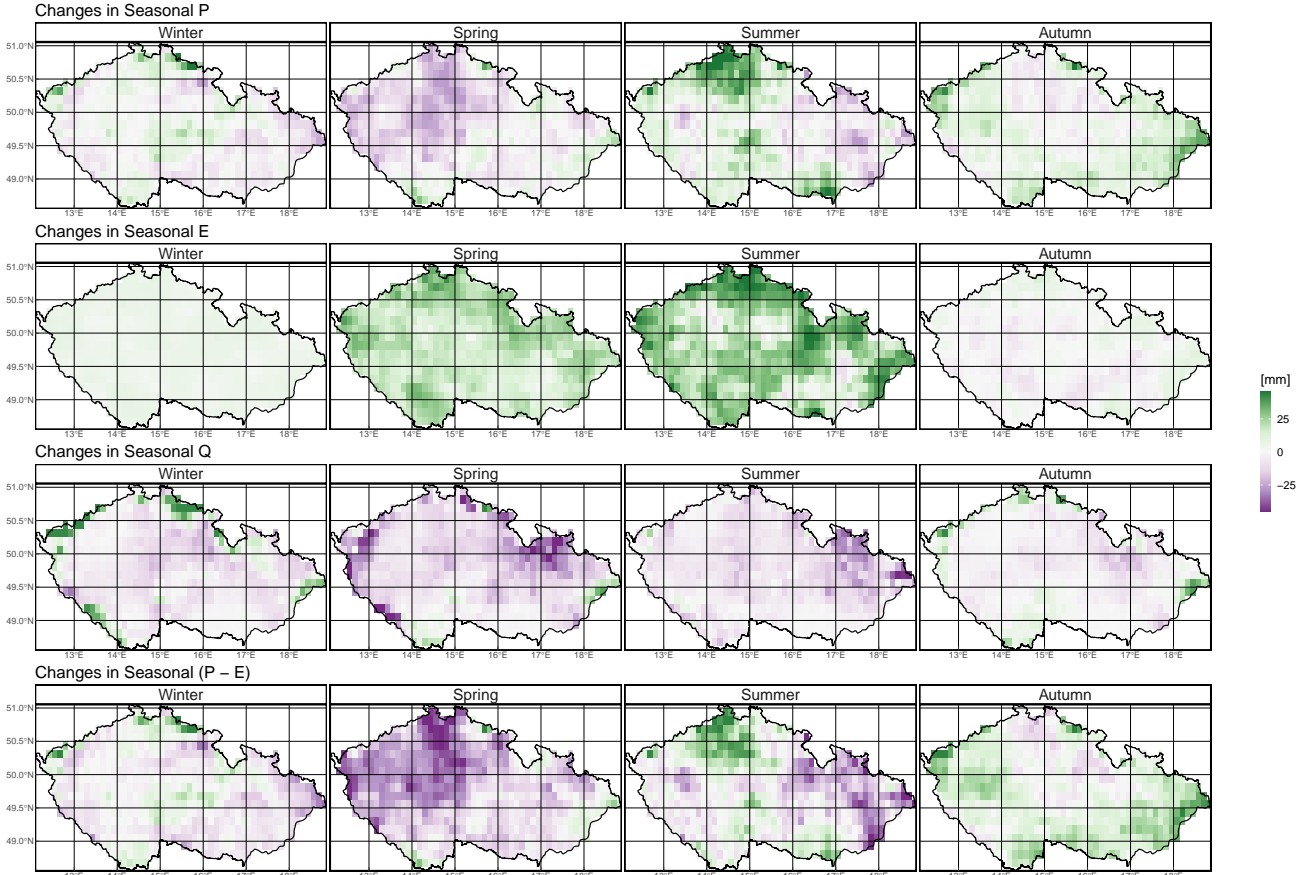

**Figure 8.** mHM spatial pattern of changes in seasonal median water fluxes over Czechia between two 30-year periods: 1961-1990 and 1991-

2020. I.e., the value of each grid cell is equal to the seasonal median value of 1991-2020 minus the seasonal median value of 1961-1990. $P$ is

precipitation, $E$ is evapotranspiration, $Q$ is runoff, and $P - E$ is precipitation minus evapotranspiration. The seasons are defined as follows:

winter as December, January, and February; spring as March, April, and May; summer as June, July, and August; autumn as September,

October, and November.

ERA5-Land spatial pattern of changes in seasonal median water fluxes closely resembles those of mHM (Figure 9). The

previously observed zonal pattern for precipitation change between the two 30-year medians seems to be driven by summer

changes. Evapotranspiration changes, unlike TerraClimate or mHM, are increasing across all seasons. With specifically large

evapotranspiration increases in summer followed by spring. In opposition, runoff has decreased regardless of the season. The

sporadic patches of increased runoff observed in mHM near the Czech borders are nonexistent in ERA5-Land. Similarly, the





mixed patterns for $P - E$ for mHM present in winter and summer are missing in ERA5-Land, which only reports decreasing changes. Lastly, we evince contrast in the direction of change between runoff (predominantly decreasing) and $P - E$ (predominantly increasing) in autumn, parallel to that of mHM. While this contrast is present in all data sets, the season differs for mHM and ERA5-Land (autumn) vs. TerraClimate (winter). Moreover, it is also inversed, i.e., TerraClimate has increasing runoff and decreasing $P - E$, but mHM and ERA5-Land have decreasing runoff and increasing $P - E$.

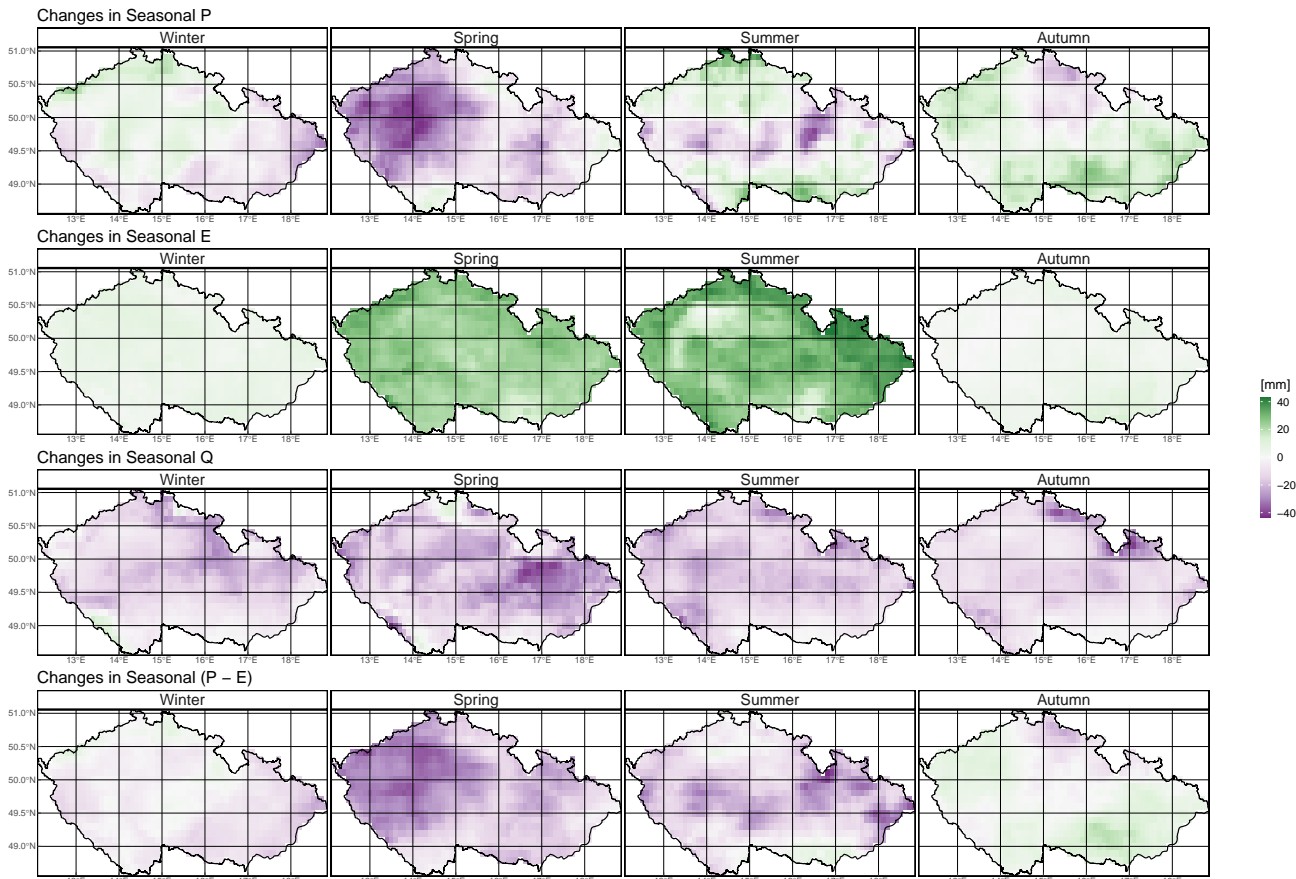

**Figure 9.** ERA5-Land spatial pattern of changes in seasonal median water fluxes over Czechia between two 30-year periods: 1961-1990 and 1991-2020. I.e., the value of each grid cell is equal to the seasonal median value of 1991-2020 minus the seasonal median value of 1961-1990. $P$ is precipitation, $E$ is evapotranspiration, $Q$ is runoff, and $P - E$ is precipitation minus evapotranspiration. The seasons are defined as follows: winter as December, January, and February; spring as March, April, and May; summer as June, July, and August; autumn as September, October, and November.





## 4 Discussion

Overall long-term changes in the annual water cycle in Czechia are primarily evident in evapotranspiration. Interestingly, the
general agreement among different data sets at low-frequency time scales dissolves as we deepen into seasonal and monthly
scales. Higher frequency temporal analysis revealed that while its seasonality modulates changes in precipitation, these changes
are overwhelmed by a consistent evapotranspiration increase. This compound behavior results in depleted water availability, as
reflected by decreasing runoff and $P - E$. Furthermore, different data combinations estimate different spatiotemporal patterns
of water cycle changes. The observed redistribution of water availability can seriously impact water resources in the region,
including the quality and quantity of drinking water, the accessibility of water for irrigation and energy generation, and the
health of aquatic ecosystems. Our results herein provide an updated overview of the water cycle in Czechia and map changes
in the past 60 years, are essential to assess and ensure the sustainable use and management of water resources in Czechia.
Additionally, we have defined and demonstrated the ability of a purely empirical ranking method to benchmark hydroclimatic
data fusion and determine the best combination to represent water cycle budget closure that can be applied to any other regional
study.

We determined that the best data sets for long-term assessment of water cycle individual components in Czechia based on the
selected references are: mHM(E-OBS), ERA5-Land, and TerraClimate for precipitation; ERA5-Land, mHM, and TerraClimate
for evapotranspiration; mHM, TerraClimate, and ERA5-Land for runoff. Similar standings for precipitation data were reported
by Fallah et al. (2020) and Bandhauer et al. (2022). Fallah et al. (2020) used runoff simulation vs. streamflow observations
using different data sets to benchmark precipitation data and found that E-OBS yields a robust agreement, while ERA5, Global
Precipitation Climatology Centre (GPCC V.2018; Schneider et al., 2011), and Multi-Source Weighted-Ensemble Precipitation
(MSWEP V2; Beck et al., 2019) show good performances. Bandhauer et al. (2022) report that while E-OBS and ERA5 agree
qualitatively, ERA5 considerably overestimates mean precipitation over Europe due to too many wet days. These prevalent wet
bias in ERA5 has been reported along diverse assessments (e.g., Bešťáková et al., 2022; Lavers et al., 2022). NCEP/NCAR R1
had the worst precipitation performance. It was previously reported that, at least regarding extreme precipitation, NCEP/NCAR
R1 performed far better than ERA5's predecessors, i.e., ERA40 (Uppala et al., 2005) and ERA-Interim (Dee et al., 2011), (Sun
et al., 2018). This disagreement could be attributed to the improvements implemented in ERA5 over its predecessors in model
parameterizations, spatial resolution, and input data assimilation. Additionally, the poor performance of NCEP/NCAR R1
might be rooted in its coarse spatial resolution (two grid cells cover Czechia).

Regarding evapotranspiration estimates, ERA5-Land has been reported as an adequate data source to overcome the unavail-
ability of observed agrometeorological data in Europe (Vanella et al., 2022), and its robustness supports its use for drought
monitoring (Vicente-Serrano et al., 2022). mHM has undergone extensive evaluation over Europe at multiple spatial scales and
has repeatedly shown its ability to capture the observed dynamics of actual evapotranspiration (Hanel et al., 2018; Rakovec
et al., 2016a) and its application to determine dominant drought types and their evolution (Markonis et al., 2021). While, to our
knowledge, there have not been studies focusing on the quality or applications of TerraClimate evapotranspiration to date, it
has been calibrated and validated using FLUXNET data (Abatzoglou et al., 2018), a conglomerate of networks gathering and





standardizing quality control protocols for station-based evapotranspiration measurements (Pastorello et al., 2020). Most of the abovementioned referenced studies also testify to the quality of runoff data from mHM, TerraClimate, and ERA5-Land because the studies use runoff and streamflow data derived, among other variables, from their evapotranspiration estimates and show

that they can capture the streamflow dynamics adequately across a wide range of climate and physiographical characteristics.

Despite our evaluation of individual water cycle components being cohesive with previous literature and even though mHM's performance was among the best for all water cycle components, the best data set combination ranking is actually Terraclimate exclusive (i.e., all flux estimates from the same data set). Further, to our surprise, we found that throughout our analysis, a hydrological model and reanalysis (mHM and ERA5-Land) presented more compatible spatiotemporal patterns than the

two hydrological models (mHM and TerraClimate). In terms of water cycle fluxes' magnitude, we report significant ERA5-Land overestimation of precipitation and evapotranspiration, which are in line with previously reported overestimations of summer precipitation over Central Europe (Hassler and Lauer, 2021; Rivoire et al., 2022). Regarding hydrological models, their evapotranspiration response is strongly linked to how they represent soil moisture and radiative energy at the surface (Boé and Terray, 2008; Zhao et al., 2013), leading to the visible discrepancies among mHM and TerraClimate.

There is agreement among the best-ranked data set combinations that most of the significant changes in Czech water fluxes are localized in spring, particularly in April and May. Notwithstanding, we observe that it is the summer season whose changes determine the spatiotemporal patterns of change between the 1991-2020 and 1961-1990 medians. Declining precipitation and increasing evapotranspiration in spring support reported drying trends over Czechia (Brázdil et al., 2015). In addition to these general patterns, we identified localized increases in winter runoff coupled with decreases and shifts in spring runoff around

the Sudetic, Šumava, and Ore Mountains. These changes in mountainous runoff have been previously identified and attributed to decreasing snow cover and earlier snowmelt season (Nedelcev and Jenicek, 2021), which in some Czech catchments also derive in summer low flows (Jenicek and Ledvinka, 2020). Similar seasonal developments of the snow effect on runoff have been reported over multiple mountainous catchments across the world (Berghuijs et al., 2014; Dierauer et al., 2018; Muelchi et al., 2021). Hänsel et al. (2019) remark that seasonal trends are sensitive to shifts in the season definition by one month, which

aligns with our monthly analysis because we identified significant changes in months like May and November (peripheral months of spring and autumn as defined herein). Additionally, it could be the reason behind summer, the contiguous season, dominating the long-term precipitation pattern.

The drying regime we report in Czechia, due to the gradual increase in atmospheric evaporative demand over the last 60 years (1961-2020) extends in time and space over central and eastern Europe (Bešťáková et al., 2022). Jaagus et al. (2022)

reported long-term drying trends for the 1949-2018 period in Slovakia, Hungary, Romania, Moldova, southern Poland, and particularly significant in Czechia. Trnka et al. (2016) described a strong tendency towards increased dryness in most Central Europe. Brázdil et al. (2009) performed one of the longest-record analysis in the region (1881-2006) and exposed an increasing tendency towards more prolonged and more intensive dry episodes. Still, it remains unclear how this long-term shift is linked to the post-2000 seasonal (Potopová et al., 2015), annual (Hanel et al., 2018), and multi-year droughts (Moravec et al., 2021)

that have occurred in Central Europe and Czechia in specific. It has been demonstrated, though, that these droughts manifest more as soil moisture deficits than meteorological and hydrological droughts, as they are related to high evaporative demand



during the warm season period (Markonis et al., 2021). Our results agreement shows that the long-term aridification could be the outcome of the same physical mechanism, i.e., evaporation increase, to the one that dominates the short-term extreme events.

Our study comes with certain limitations that pave the way for future research. A certain limitation is that our analyses do not attribute the observed changes to any potential physical or anthropogenic drivers. It is likely that the evapotranspiration increase is linked to long-term changes in atmospheric circulation patterns that have caused a decline in cloudiness (Lhotka et al., 2020). As it has been shown that global warming is going to disrupt the terrestrial water cycle mainly due to changes in precipitation (Roderick et al., 2014), it is more plausible to attribute the observed intensification to the fluctuations of

atmospheric circulation. Yet, this remains to be confirmed by future studies that will determine the factors that contribute most to the hydroclimatic shifts, although drought projections over Czechia (Dubrovsky et al., 2009), and central Europe Hari et al. (2020) indicate an increased drought risk in the future prevalent under different climate change scenarios. Additionally, our work does not investigate the role of water storage (snow and groundwater), as well as land cover or vegetation changes. Lastly, while country-level assessments are essential to improve water resources management and natural hazard policies, the water

cycle budget is closed over hydrological units, not administrative boundaries.

## 5    Conclusions

Herein, we have proposed and demonstrated the applicability of a novel benchmarking method based on water cycle budget closure for hydroclimatic data fusion. The method does not enforce closure nor merge multiple data sets into a new one, but instead identifies the best combination of data sets in terms of water cycle budget residual distribution and correlation

to referential data. Furthermore, the ranking method presented could easily be applied to any other region and use different referential data sets for evaluation. The ranking method may still be employed using gridded data like GPCC or CRU TS as an evaluation reference in data-scarce areas or when ground-station data is not publicly available. Most importantly, this metric is not constrained by data availability, as any of the variables in the equation evaluation terms can be omitted. This modularity makes it a flexible alternative to traditional approaches.

Using the best water budget data, we demonstrate that Czechia is undergoing water cycle acceleration, evinced by increased atmospheric water demand. Remarkably, the increase in precipitation is not as pronounced as that one in evapotranspiration. While changes in the 30-year median of spatial weight average annual values show a minimum change in water availability, the spatial patterns reveal a prevalent decreasing pattern of runoff across the country. Besides, we identified significant spatial heterogeneity when assessing precipitation at a seasonal scale. Intriguingly, summer patterns are reflected in the spatial differ-

ence between the 1991-2020 and the 1961-1990 medians despite most of the significant changes in water cycle components being localized in spring. What is more, the precipitation rain/snow partition effect of less snow and earlier snowmelt around the mountains is reflected in a seasonal shift of runoff (increase in winter and subsequent decrease in spring). This might reflect how sub-seasonal shifts could affect the long-term hydrologic changes.





Based on our results and previous literature, it is safe to state that the depletion of water availability (runoff and $P - E$)
over Czechia could prompt a surge in drought frequency. Considering that shifts in evapotranspiration overwhelm those of
precipitation, the water cycle in Czechia is mainly driven by changes in energy rather than water availability. Further research
is needed to better understand the complex drivers of this drying trend and to develop targeted interventions to address possible
factors external to natural variability, like land-use changes and other anthropogenic factors. Although it remains unknown if
this drying trend will persist, it should be considered in the planning of effective drought management strategies and water
conservation measures to mitigate its adverse impacts for agriculture, energy production, and natural ecosystems in Czechia.

*Code and data availability.* The data compiled herein and the R code for the figures are publicly available at https://github.com/MiRoVaGo/
ugc_cwc.

*Author contributions.* Mijael Rodrigo Vargas Godoy: Conceptualization, Formal analysis, Investigation, Writing - Original Draft. Yannis
Markonis: Conceptualization, Supervision, Writing - Review & Editing. Oldrich Rakovec: Investigation, Writing - Review & Editing. Michal
Jenicek: Investigation, Writing - Review & Editing. Riya Dutta: Writing - Review & Editing. Rajani Kumar Pradhan: Investigation. Zuzana
Bešťáková: Investigation. Jan Kyselý: Writing - Review & Editing. Roman Juras: Writing - Review & Editing. Simon Michael Papalexiou:
Writing - Review & Editing. Martin Hanel: Writing - Review & Editing.

*Competing interests.* The authors declare that they have no conflict of interest.

*Acknowledgements.* This work was carried out within the project "Investigation of Terrestrial HydrologicAl Cycle Acceleration (ITHACA)"
funded by the Czech Science Foundation (Grant 22-33266M). MRVG was supported by the project "Water Cycle Intensification Over the
Czech Republic" from the University Grant Competition (UGC No. 65/2021), under the Improvement in Quality of the Internal Grant Scheme
at CZU, reg. no. CZ.02.2.69/0.0/0.0/19_073/0016944.



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
