# Peer review of "Water Cycle Changes in Czechia: A Multi-Source Water Budget Perspective"

_Hydrology and Earth System Sciences, 2023_

## Author Comment (AC1)

**Reply to Reviewer 1**

The study presents a statistical analysis of the hydrological cycle in Czechia. To do so the study uses multiple gridded hydrological products, derived using remote sensing and reanalysis. First a ranking scheme regarding the performance of each product and their combination is presented. To me this is the main novelty of the study. Afterwards the best products are analyzed to provide spatially explicit estimates of the change of hydrological dynamics between a past and a present era.

We would like to thank the reviewer for their brief yet insightful comments. We have revised the manuscript based on the reviewer's comments and suggestions. The evaluation data sets will be replaced by high-quality observations only. In addition, we will add new figures to present further results and discuss their implications supporting the hypothesis of re-distribution of terrestrial water, since Czechia is losing water in the long-term (precipitation remains the same while evapotranspiration increases). In the following, we provide detailed replies to all comments and discuss changes to the main manuscript.

Overall, the methodology is mostly solid. My main methodological question concerns the use of GLEAM and GRUN. I am not sure why GLEAM and GRUN were used as benchmark datasets. GLEAM and GRUN are both model based. How can they be used as benchmarks for validation? They themselves carry a lot of uncertainty. For ET, the physical basis of some of the remaining datasets (e.g., ERA5 land) is much more detailed than GLEAM as they integrate a full complexity land surface scheme, rather than simplifying models (e.g., Priestley Taylor). GRUN has even less physical basis, as it is a statistical model. I would be more convinced with the analysis, if only real high-quality observations were included in benchmarking the various datasets.

Initially GLEAM and GRUN were chosen as evaluation benchmarks because both are considered high quality products (E.g., Yang et al. [2017]; Bai and Liu [2018]; Liu et al. [2021]; Hu et al. [2021]; Xiong et al. [2022]; Xu et al. [2022]; Mei et al. [2023]) but their record lengths were not long enough to be part of the main analysis. As correctly pointed out by the reviewer, these data sets do carry considerable uncertainty. Therefore, in order to include only high-quality observational data and to evince the robustness of the ranking method proposed we decided to replace GRUN by GRDC for runoff and perform the ranking without an evapotranspiration reference for evaluation. Note that we selected only three stations from GRDC, namely the Bohumin (Oder), Decin (Elbe), and Moravsky Jan (Danube) stations, which are placed near the borders of the country and their wieghted average was computed using the catchment area as registered by GRDC. The revised benchmarking (revised Figure 2) and top ranking results vary only slightly (revised Table 2), further supporting our initial choice of referential data sets.

[Figure]

**Figure 2.** Benchmarking spatial weighted average annual water fluxes over Czechia between 1961 and 2020. For consistency and comparability between different water fluxes, annual anomalies were computed using the 1981-2010 average as a reference, the common period among all data sets. The 1981-2010 average and standard deviation are listed at the bottom left of each panel. Linear correlation summary statistics are displayed at the bottom right of each panel. The spread of the estimates being evaluated is shown in gray, and their mean is in white. (a) Precipitation evaluation. CHMI data is shown in blue. (b) Evapotranspiration evaluation. (c) Runoff evaluation. GRDC (Bohumin, Decin, and Moravsky Jan stations) data is shown in purple.

**Table 2.** Data set ranking as determined by Equation 3. $P$ is precipitation, $E$ is evapotranspiration, $Q$ is runoff, $\bar{\xi}$ is the mean residual over 60 years, $\sigma_\xi$ is the standard deviation of the residual over 60 years, $cor(P-E,Q)$ is the correlation between $P-E$ and $Q$ for the $i$-th ranked combination, $cor(P,P_o)$ is the correlation between $P$ of the $i$-th ranked combination and CHMI, and $cor(Q,Q_o)$ is the correlation between $Q$ of the $i$-th ranked combination and GRDC.

| Ranking | $P$ | $E$ | $Q$ | $\bar{\xi}$ | $\sigma_\xi$ | $cor(P-E,Q)$ | $cor(P,P_o)$ | $cor(Q,Q_o)$ |
|---|---|---|---|---|---|---|---|---|
| 1st | TerraClimate | TerraClimate | TerraClimate | -0.346 | 30.204 | 0.846 | 0.941 | 0.836 |
| 2nd | mHM(E-OBS) | mHM | mHM | -0.912 | 51.231 | 0.816 | 0.994 | 0.967 |
| 3rd | CRU TS v4.06 | TerraClimate | TerraClimate | -1.749 | 29.944 | 0.843 | 0.938 | 0.836 |
| 4th | TerraClimate | TerraClimate | mHM | -8.861 | 39.847 | 0.730 | 0.941 | 0.967 |
| 5th | CRU TS v4.06 | TerraClimate | mHM | -10.265 | 40.613 | 0.711 | 0.938 | 0.967 |
| 6th | ERA5-Land | ERA5-Land | ERA5-Land | -5.554 | 66.606 | 0.701 | 0.951 | 0.882 |
| ... | | | | ... | ... | ... | ... | ... |
| 14th | PRECL/L | mHM | TerraClimate | 17.013 | 60.281 | 0.658 | 0.891 | 0.836 |
| ... | | | | ... | ... | ... | ... | ... |
| 24th | ERA5-Land | TerraClimate | mHM | 114.628 | 44.721 | 0.763 | 0.951 | 0.967 |
| ... | | | | ... | ... | ... | ... | ... |
| 38th | ERA5-Land | NCEP/NCAR R1 | mHM | -166.746 | 60.420 | 0.714 | 0.951 | 0.967 |
| ... | | | | ... | ... | ... | ... | ... |
| 48th | PREC/L | mHM | ERA5-Land | -52.549 | 82.751 | 0.382 | 0.891 | 0.882 |
| ... | | | | ... | ... | ... | ... | ... |
| 72nd | mHM(E-OBS) | mHM | NCEP/NCAR R1 | -134.044 | 87.923 | 0.237 | 0.994 | 0.405 |
| ... | | | | ... | ... | ... | ... | ... |
| 87th | NCEP/NCAR R1 | NCEP/NCAR R1 | NCEP/NCAR R1 | -292.024 | 137.297 | 0.675 | 0.181 | 0.405 |
| ... | | | | ... | ... | ... | ... | ... |
| 96th | CRU TS v4.06 | NCEP/NCAR R1 | NCEP/NCAR R1 | -424.772 | 93.962 | -0.019 | 0.938 | 0.405 |

Apart from that, a thorough analysis is presented, which to a large extent is consistent with previous results related to continental Europe. Even though the study is methodologically sound, its novelty is limited in my opinion because of (a) the data products used are all well established and have been extensively previously analyzed at regional and global scales, and (b) the limited geographical extent of the study. I find the paper better suited to journals focusing on regional studies, rather than HESS whose goal is to further advance the fundamental understanding of hydrological processes and their impacts on society and ecosystems.

We thank the reviewer for this comment because it helped us realize that the novelties of our study have not been properly highlighted. Although the data products have been previously analyzed at regional or global scales, this is done under a univariate perspective, that does not consider the ability of the data sets to reproduce the water cycle (and its changes) as a whole in a structurally plausible manner. This comment pushed us to look deeper into the water budget closure, where it became evident that there is a substantial overestimation of the drying in ERA5-Land (Figure 3).

[Figure]

**Figure 3.** Spatial weighted average annual water fluxes over Czechia (first row), Danube basin inside Czechia (second row), Elbe basin inside Czechia (third row), and Oder basin inside Czechia (fourth row). Where $P$ is precipitation in blue, $E$ is evapotranspiration in green, $Q$ is runoff in purple, $\xi$ is the residual $(P - E - Q)$ in black, and cumsum($\xi$) is the cumulative sum of the residual in orange. Left column: TerraClimate ($P$), TerraClimate ($E$), and TerraClimate ($Q$). Middle column: mHM(E-OBS) ($P$), mHM ($E$), and mHM ($Q$). Right column: ERA5-Land ($P$), ERA5-Land ($E$), and ERA5-Land ($Q$).

We acknowledge that the geographical extent of the study is small. Nonetheless, a relatively small study domain is not uncommon at HESS, as demonstrated by some of the work we cited [Jenicek and Ledvinka, 2020; Muelchi et al., 2021]. The former discusses the influence of snow storage and snowmelt inter-annual variations effect on seasonal runoff and summer low flows in *Czechia*. The latter addresses projected changes in river runoff regimes in *Switzerland*. Furthermore, there are multiple other publications in recent years with similar geographical extents, some of which are:

- Osuch et al. [2016] Reported possible climate change effects on dryness by assessing the standardized precipitation index on multiple climate projections in *Poland*.
- Silvestro et al. [2018] Analyzed streamflow extremes and long-term water balance in the *Liguria region of Italy*.
- Girons Lopez et al. [2021] Benchmarked the SHYPE operational hydrological model in *Sweden*.
- Hanus et al. [2021] Reported changes in runoff signatures at multiple scales in contrasting Alpine catchments in *Austria*.
- Torelló-Sentelles and Franzke [2022] Presented a random forest model to predict drought impacts in *Spain*.
- Alexopoulos et al. [2023] Evaluated precipitation reanalyses performance for rainfall-runoff modeling using the GR4H model in *Slovenia*

Despite their regional geographical extent, the findings of the above-mentioned have implications to our understanding about the hydrological processes. Likewise, in the revised version we will highlight the main novelties of our study, which is the importance of combining data sources that describe all the components of the terrestrial water cycle and presenting a showcase of inconsistencies that might not be visible if the single components are evaluated as performance metrics (the case of ERA5-Land).

**A few minor comments:**

Lines 17-20: Not clear what the contradiction is between the 2 statements

For clarity and brevity the text will be rephrased from: "On the one hand, small changes in total precipitation suggest a shift in precipitation towards more intense and less frequent events [Trenberth, 2011]. On the other hand, it was hypothesized that an increased vertical gradient of atmospheric water vapor would offset atmospheric wind convergence in the tropics making wet regions wetter and dry regions drier [Held and Soden, 2006]."

To: "It was hypothesized that an increased vertical gradient of atmospheric water vapor would offset atmospheric wind convergence in the tropics making wet regions wetter and dry regions drier [Held and Soden, 2006]."

Line 26: define what you mean by unquantified uncertainties

For clarification the text will be rephrased from: "... unquantified uncertainties on satellite-based products [Sheffield et al., 2009]."

To: "... unquantified uncertainties on satellite-based products [e.g., the impact of cloud filtering; Povey and Grainger, 2015]."

Line 73: What is the meaning of the roof analogy?

For clarity and brevity, we will remove the roof analogy. Which was meant to be a literary figure for a headwaters region (water falling on top primarily runs away rather than staying in).

Figure 1: the different shading is not clear. I suggest the authors to add in bold colors the catchment boundaries for clarity

Figure 1 will be updated as suggested.

[Figure]

**Figure 1.** The three drainage basins within Czechia's boundaries. Elbe (light gray shade), Danube (striped dark gray shade), and Oder (no shade).

Line 130: System instead of set of ODEs

Text will be replaced.

I find the definitions of R2 and RMSE a bit redundant.

The definitions will be removed.

In eq 1, 2 I suggest changing the variable name of the residual term from R to something different, e.g. epsilon, to not confuse the reader as R is commonly used for runoff, and previously in the paper as the coefficient of determination

To avoid confusion the variable name will be changed from $R$ to $\xi$.

Line 211: Why were the authors surprised by the quality of ERA5-Land. Please explain further this statement? The land surface scheme of ERA5-Land (H-TESSEL) has a hydrological component, which is in compatible complexity with the remaining hydrological models of the study.

To explicitly refer to the cause of surprise the text will be rephrased from : "Notwithstanding, we were surprised to see the ERA5-Land exclusive combination (i.e., all flux estimates from the same data set) among the top five ranks."

To: "Notwithstanding, we were surprised to see the ERA5-Land exclusive combination (i.e., all flux estimates from the same data set) among the top six ranks despite non steady water budget residuals (Figure 3) as well as biases 1.7-3.3 and 3.8-4.2 times larger than those of models for runoff (Figure 2c) and precipitation (Figure 2a), respectively" As the previous sentence states: "We expected combinations with hydrological model data to be highly ranked and reanalyses to be poorly ranked due to the above-reported considerable biases of the latter."

[Figure]

**Figure 3.** Spatial weighted average annual water fluxes over Czechia (first row), Danube basin inside Czechia (second row), Elbe basin inside Czechia (third row), and Oder basin inside Czechia (fourth row). Where $P$ is precipitation in blue, $E$ is evapotranspiration in green, $Q$ is runoff in purple, $\xi$ is the residual $(P - E - Q)$ in black, and cumsum($\xi$) is the cumulative sum of the residual in orange. Left column: TerraClimate ($P$), TerraClimate ($E$), and TerraClimate ($Q$). Middle column: mHM(E-OBS) ($P$), mHM ($E$), and mHM ($Q$). Right column: ERA5-Land ($P$), ERA5-Land ($E$), and ERA5-Land ($Q$).

Line 225-228: Does this imply that the models do not close the water balance, or that the integration periods are not long enough, and the discrepancies are due to soil water storage dynamics?

We took 30-year periods, the minimum required to calculate a climate normal, and it would be safe to assume negligible change in water storage. Which is supported by the stationary time series seemingly centered around zero (Figure 3). Moreover, we cannot assert that models do not close the water balance because the discrepancies are considerably small compared to the values of those fluxes.

Line 244: Change Abril to April

Text will be changed.

Figure 4: Might be better if presented as cumulative distribution functions, q-q plots or boxplots.

Figure 4 will be revised from a histogram to a boxplot. Please note that the revised figure numbering is now Figure 5 due to the newly added Figure 3.

[Figure]

**Figure 5.** Box plots of spatial weighted average annual water fluxes over Czechia, where *P* is precipitation, *E* is evapotranspiration, *Q* is runoff, and *P* − *E* is precipitation minus evapotranspiration. Data are divided into two 30-year periods: 1961-1990 (blue) and 1991-2020 (yellow). Note that outliers are present only in the latter period (i.e., 1991-2020) as expected from the recorded severe drought of 2003.

**References**

Alexopoulos MJ, Müller-Thomy H, Nistahl P, Šraj M, Bezak N (2023) Validation of precipitation reanalysis products for rainfall-runoff modelling in Slovenia. Hydrology and Earth System Sciences 27(13):2559–2578, DOI 10.5194/hess-27-2559-2023, publisher: Copernicus GmbH

Bai P, Liu X (2018) Intercomparison and evaluation of three global high-resolution evapotranspiration products across China. Journal of Hydrology 566:743–755, DOI 10.1016/j.jhydrol.2018.09.065

Girons Lopez M, Crochemore L, Pechlivanidis IG (2021) Benchmarking an operational hydrological model for providing seasonal forecasts in Sweden. Hydrology and Earth System Sciences 25(3):1189–1209, DOI 10.5194/hess-25-1189-2021, publisher: Copernicus GmbH

Hanus S, Hrachowitz M, Zekollari H, Schoups G, Vizcaino M, Kaitna R (2021) Future changes in annual, seasonal and monthly runoff signatures in contrasting Alpine catchments in Austria. Hydrology and Earth System Sciences 25(6):3429–3453, DOI 10.5194/hess-25-3429-2021, publisher: Copernicus GmbH

Held IM, Soden BJ (2006) Robust Responses of the Hydrological Cycle to Global Warming. Journal of Climate 19(21):5686–5699, DOI 10.1175/JCLI3990.1, publisher: American Meteorological Society Section: Journal of Climate

Hu Y, Duan W, Chen Y, Zou S, Kayumba PM, Sahu N (2021) An integrated assessment of runoff dynamics in the Amu Darya River Basin: Confronting climate change and multiple human activities, 1960–2017. Journal of Hydrology 603:126905, DOI 10.1016/j.jhydrol.2021.126905

Jenicek M, Ledvinka O (2020) Importance of snowmelt contribution to seasonal runoff and summer low flows in Czechia. Hydrology and Earth System Sciences 24(7):3475–3491, DOI 10.5194/hess-24-3475-2020, publisher: Copernicus GmbH

Liu J, Zhang J, Kong D, Feng X, Feng S, Xiao M (2021) Contributions of Anthropogenic Forcings to Evapotranspiration Changes Over 1980–2020 Using GLEAM and CMIP6 Simulations. Journal of Geophysical Research: Atmospheres 126(22):e2021JD035367, DOI 10.1029/2021JD035367, _eprint: https://onlinelibrary.wiley.com/doi/pdf/10.1029/2021JD035367

Mei Y, Mai J, Do HX, Gronewold A, Reeves H, Eberts S, Niswonger R, Regan RS, Hunt RJ (2023) Can Hydrological Models Benefit From Using Global Soil Moisture, Evapotranspiration, and Runoff Products as Calibration Targets? Water Resources Research 59(2):e2022WR032064, DOI 10.1029/2022WR032064, URL https://onlinelibrary.wiley.com/doi/abs/10.1029/2022WR032064, _eprint: https://onlinelibrary.wiley.com/doi/pdf/10.1029/2022WR032064

Muelchi R, Rössler O, Schwanbeck J, Weingartner R, Martius O (2021) River runoff in Switzerland in a changing climate – runoff regime changes and their time of emergence. Hydrology and Earth System Sciences 25(6):3071–3086, DOI 10.5194/hess-25-3071-2021, URL https://hess.copernicus.org/articles/25/3071/2021/, publisher: Copernicus GmbH

Osuch M, Romanowicz RJ, Lawrence D, Wong WK (2016) Trends in projections of standardized precipitation indices in a future climate in Poland. Hydrology and Earth System Sciences 20(5):1947–1969, DOI 10.5194/hess-20-1947-2016, publisher: Copernicus GmbH

Povey AC, Grainger RG (2015) Known and unknown unknowns: uncertainty estimation in satellite remote sensing. Atmospheric Measurement Techniques 8(11):4699–4718, DOI 10.5194/amt-8-4699-2015, publisher: Copernicus GmbH

Sheffield J, Ferguson CR, Troy TJ, Wood EF, McCabe MF (2009) Closing the terrestrial water budget from satellite remote sensing. Geophysical Research Letters 36(7), publisher: Wiley Online Library

Silvestro F, Parodi A, Campo L, Ferraris L (2018) Analysis of the streamflow extremes and long-term water balance in the Liguria region of Italy using a cloud-permitting grid spacing reanalysis dataset. Hydrology and Earth System Sciences 22(10):5403–5426, DOI 10.5194/hess-22-5403-2018, publisher: Copernicus GmbH

Torelló-Sentelles H, Franzke CLE (2022) Drought impact links to meteorological drought indicators and predictability in Spain. Hydrology and Earth System Sciences 26(7):1821–1844, DOI 10.5194/hess-26-1821-2022, publisher: Copernicus GmbH

Trenberth KE (2011) Changes in precipitation with climate change. Climate Research 47(1-2):123–138

Xiong J, Yin J, Guo S, He S, Chen J, Abhishek (2022) Annual runoff coefficient variation in a changing environment: a global perspective. Environmental Research Letters 17(6):064006, DOI 10.1088/1748-9326/ac62ad, URL https://dx.doi.org/10.1088/1748-9326/ac62ad, publisher: IOP Publishing

Xu D, Bisht G, Sargsyan K, Liao C, Leung LR (2022) Using a surrogate-assisted Bayesian framework to calibrate the runoff-generation scheme in the Energy Exascale Earth System Model (E3SM) v1. Geoscientific Model Development 15(12):5021–5043, DOI 10.5194/gmd-15-5021-2022, publisher: Copernicus GmbH

Yang X, Yong B, Ren L, Zhang Y, Long D (2017) Multi-scale validation of GLEAM evapotranspiration products over China via ChinaFLUX ET measurements. International Journal of Remote Sensing 38(20):5688–5709, DOI 10.1080/01431161.2017.1346400, publisher: Taylor & Francis _eprint: https://doi.org/10.1080/01431161.2017.1346400

---

## Author Comment (AC2)

**Reply to Reviewer 2**

This article presents extensive work on comparing the performance of different datasets on the closure degree of the water budget and demonstrates the acceleration in the hydrological cycle over Czechia. Overall, the paper is well written and readable and provides direct evidence of the performance on evaluation from different datasets. However, the title of the article could probably be rephrased, as it looks like a new method for demonstrating water cycle acceleration, but the actual story of the article is more about comparing the performance of different datasets using a novel method. Here are several issues needed to be addressed or clarified, which are listed as follows.

We thank the reviewer for their encouraging feedback and detailed comments. The empirical ranking framework we propose is our original approach. However, the focus of our work is to assess how different data sets portray different stories. While often there are similarities between data sets specially at coarser scales, in reality, each data sets depicts a different scenario. We will change the manuscript title from: "Water Cycle Acceleration in Czechia: A Water Budget Approach"

To: "Water Cycle Changes in Czechia: A Multi-Source Water Budget Perspective"

In the following, we provide detailed replies to all comments and discuss changes to the main manuscript.

**Major comments:**

Line 12: What does the median space pattern mean here? Why only mention spring and summer here?

To add further detail and clarity the text will be rephrased from: "Interestingly, the most significant temporal changes in Czechia take place during spring, while median spatial patterns stem from summer changes in the water cycle."

To: "Interestingly, the most significant temporal changes in Czechia occur during spring, while the spatial pattern of the change in median values stems from summer changes in the water cycle, which are the seasons within the months with statistically significant changes."

Line 17-21: A more logical organization is needed, perhaps adding a sentence in front of "on the one hand" to introduce the relationship between the water cycle and water fluxes you have chosen here (precipitation, evapotranspiration. . . ). The information behind "on the one hand" and "on the other hand" are not parallel associations, and these two aspects are less relevant to the focus of this article.

For clarity and brevity the text will be rephrased from: "On the one hand, small changes in total precipitation suggest a shift in precipitation towards more intense and less frequent events [Trenberth, 2011]. On the other hand, it was hypothesized that an increased vertical gradient of atmospheric water vapor would offset atmospheric wind convergence in the tropics making wet regions wetter and dry regions drier [Held and Soden, 2006]."

To: "It was hypothesized that an increased vertical gradient of atmospheric water vapor would offset atmospheric wind convergence in the tropics making wet regions wetter and dry regions drier [Held and Soden, 2006]."

Line 36-43: The information in parentheses may be summarized in a supplementary table and moved the table to supplementary materials for detailed clarification. In addition, please add the datasets categories (which ones belong to satellites or ground-based measurements, or climate models) in the table.

As suggested the information will be added as supplementary tables (Table S1, S2, S3, S4, and S5) in a formatting compatible with revised Table 1.

**Table S1.** Compiled from Sahoo et al. [2011]. $P$ is precipitation, $E$ is evapotranspiration, $Q$ is runoff, and $\Delta$ TWS is changes in total water storage.

| Name | Variable | Spatial Resolution | Temporal Resolution | Record Length | Data Type | Reference(s) |
|---|---|---|---|---|---|---|
| GPCP | $P$ | 1° | Daily | 1997-2006 | Satellite-based | Adler et al. [2003] |
| TMPA 3B42RT | $P$ | 0.25° | 3h | 1997-2019 | Satellite-based | Huffman et al. [2007] |
| CMORPH | $P$ | 8km | 30min | 2003-2006 | Satellite-based | Joyce et al. [2004] |
| PERSIANN | $P$ | 0.25° | 3h | 2000-2006 | Satellite-based | Hong et al. [2004] |
| CPC PRECL | $P$ | 2.5° | Monthly | 1950-Present | Gauge-based | Chen et al. [2002] |
| CRU TS3.0 | $P$ | 0.5° | Monthly | 1901-2006 | Gauge-based | Mitchell and Jones [2005] |
| WM v2.01 | $P$ | 0.5° | Monthly | 1900-2008 | Gauge-based | Willmott and Matsuura [2001] |
| GPCC | $P$ | 0.5° | Monthly | 1900-2007 | Gauge-based | Schneider et al. [2011] |
| PM (ISCCP) | $E$ | 2.5° | 3h | 1984-2005 | Satellite-based | Sheffield et al. [2010] |
| PM (EOS) | $E$ | 5km | Daily | 2003-2006 | Satellite-based | Vinukollu et al. [2011] |
| PT (EOS) | $E$ | 5km | Daily | 2003-2006 | Satellite-based | Vinukollu et al. [2011] |
| SEBS (EOS) | $E$ | 5km | Daily | 2003-2006 | Satellite-based | Vinukollu et al. [2011] |
| VIC | $E$ | 1.0° | 3h | 1948-2006 | Model | Sheffield and Wood [2007] |
| ERA-interim | $E$ | T255 | 12h | 1989-2006 | Reanalysis | Simmons [2006] |
| GRACE | $\Delta$ TWS | Basin | ~Monthly | 2002-2006 | Satellite-based | Swenson and Wahr [2006] |
| GRDC | $Q$ | Basin | Monthly | 1900-2006 | Station | www.bafg.de/GRDC |

**Table S2.** Compiled from Pan et al. [2012]. $P$ is precipitation, $E$ is evapotranspiration, $Q$ is runoff, and $\Delta$ TWS is changes in total water storage.

| Name | Variable | Spatial Resolution | Temporal Resolution | Record Length | Data Type | Reference(s) |
|---|---|---|---|---|---|---|
| GPCP v2.2 | $P$ | 2.5° | Monthly | 1950-Present | Gauge-based | Adler et al. [2003] |
| CRU TS3.0 | $P$ | 0.5° | Monthly | 1901-2006 | Gauge-based | Mitchell and Jones [2005] |
| WM v2.01 | $P$ | 0.5° | Monthly | 1900-2008 | Gauge-based | Willmott and Matsuura [2001] |
| GPCC | $P$ | 0.5° | Monthly | 1900-2007 | Gauge-based | Schneider et al. [2011] |
| MPI | $E$ | 0.5° | Monthly | 1982–2008 | Flux tower-based | Jung et al. [2010] |
| SEBS (EOS) | $E$ | 5km | Daily | 2003-2006 | Satellite-based | Vinukollu et al. [2011] |
| GRACE | $\Delta$ TWS | Basin | ~Monthly | 2002-2006 | Satellite-based | Swenson and Wahr [2006] |
| GRDC | $Q$ | Basin | Monthly | 1900-2006 | Station | www.bafg.de/GRDC |

**Table S3.** Compiled from Rodell et al. [2015]. $P$ is precipitation, $E$ is evapotranspiration, $Q$ is runoff, and $\Delta$ TWS is changes in total water storage.

| Name | Variable | Spatial Resolution | Temporal Resolution | Record Length | Data Type | Reference(s) |
|---|---|---|---|---|---|---|
| GPCP v2.2 | $P$ | 1° | Daily | 1997-2006 | Satellite-based | Adler et al. [2003]; Huffman et al. [2009] |
| Princeton ET | $E$ | 5km | Daily | 2003-2006 | Satellite-based | Vinukollu et al. [2011] |
| MERRA and MERRA-Land | $E$ | 0.5°x0.667° | Hourly | 1980-2016 | Reanalysis | Rienecker et al. [2011]; Bosilovich et al. [2011]; Reichle |
| GLDAS | $E$ | 0.25° | 3h | 1948-2014 | Model | Roderick et al. [2014] |
| University of Washington runoff | $Q$ | 2° | Monthly | 1998–2008 | Model | Jung et al. [2010] |
| GRACE | $\Delta$ TWS | Basin | ~Monthly | 2002-2006 | Satellite-based | Swenson and Wahr [2006] |

**Table S4.** Compiled from Zhang et al. [2016]. $P$ is precipitation, $E$ is evapotranspiration, $Q$ is runoff, and $\Delta$ TWS is changes in total water storage.

| Name | Variable | Spatial Resolution | Temporal Resolution | Record Length | Data Type | Reference(s) |
|------|----------|--------------------|---------------------|---------------|-----------|--------------|
| CSU | $P$ | 0.25° | 3h | 1998–2010 | Satellite-based | Bytheway and Kummerow [2013] |
| PGF | $P$ | 0.25° | 3h | 1948–2010 | Satellite-based | Sheffield et al. [2006] |
| CHIRPS | $P$ | 0.5° | Monthly | 1981–present | Satellite-based | Funk et al. [2014] |
| GPCC(v6) | $P$ | 0.5° | Monthly | 1901–2010 | Gauge-based | Schneider et al. [2014] |
| TMPA-RT | $P$ | 0.25° | Monthly | 2001–2019 | Satellite-based | Huffman et al. [2007, 2010] |
| SRB–PGF–PM | $E$ | 0.5° | 3h | 1984–2007 | Satellite-based | Vinukollu et al. [2011] |
| VIC | $E$ | 0.25° | 3h | 1948–2010 | Model | Sheffield and Wood [2007] |
| ERA-interim | $E$ | T255 | 12h | 1989–2006 | Reanalysis | Simmons [2006] |
| MERRA | $E$ | 0.5°x0.667° | Hourly | 1980–2016 | Reanalysis | Rienecker et al. [2011] |
| GLEAM | $E$ | 0.5° | 3h | 1984–2017 | Satellite-based | Gonzalez Miralles et al. [2011] |
| SRB-CFSR-SEBS | $E$ | 0.5° | Daily | 1984–2007 | Satellite-based | Vinukollu et al. [2011] |
| SRB-CFSR-PM | $E$ | 0.5° | Daily | 1984–2007 | Satellite-based | Vinukollu et al. [2011] |
| SRB-CFSR-PT | $E$ | 0.5° | Daily | 1984–2007 | Satellite-based | Vinukollu et al. [2011] |
| VIC | $Q$ | 0.25° | 3h | 1948–2010 | Model | Sheffield and Wood [2007] |
| VIC | $\Delta$ TWS | 0.25° | 3h | 1948–2010 | Model | Sheffield and Wood [2007] |
| GRACE | $\Delta$ TWS | 1° | Monthly | 2002–present | Satellite-based | Landerer and Swenson [2012] |

**Table S5.** Compiled from Munier and Aires [2018]. *P* is precipitation, *E* is evapotranspiration, *Q* is runoff, and ∆ TWS is changes in total water storage.

| Name | Variable | Spatial Resolution | Temporal Resolution | Record Length | Data Type | Reference(s) |
|------|----------|--------------------|--------------------|---------------|-----------|--------------|
| TMPA | *P* | 0.25° | Monthly | 1998-2019 | Satellite-based | Huffman et al. [2007] |
| CMORPH | *P* | 0.25° | Daily | 1998–present | Satellite-based | Sheffield et al. [2006] |
| NRL | *P* | 0.25° | 12h | 2003–2010 | Satellite-based | Turk et al. [2010] |
| GPCP | *P* | 2.5° | Monthly | 1979–present | Satellite-based | Schneider et al. [2014] |
| GLEAM | *E* | 0.25° | 3h | 1980–2011 | Satellite-based | Gonzalez Miralles et al. [2011] |
| MOD16 | *E* | 1km | 8-day | 2000–2012 | Satellite-based | Mu et al. [2007] |
| NTSG | *E* | 8km | Daily | 1983–2006 | Satellite-based | Zhang et al. [2010] |
| CSR | ∆ TWS | Basin | Monthly | 2002-present | Satellite-based | http://grace.jpl.nasa.gov/data/ |
| GFZ | ∆ TWS | Basin | Monthly | 2002-present | Satellite-based | http://grace.jpl.nasa.gov/data/ |
| JPL | ∆ TWS | Basin | Monthly | 2002-present | Satellite-based | http://grace.jpl.nasa.gov/data/ |
| GRGS | ∆ TWS | Basin | Monthly | 2002-present | Satellite-based | http://grgs.obs-mip.fr/grace/ |
| GRDC | *Q* | Basin | Monthly | 1900-present | Station | http://www.grdc.sr.unh.edu/ |

Table 1: Add the datasets categories (which ones belong to satellites or ground-based measurements or climate models) in table 1.

Table 1 will be revised as follows:

**Table 1.** Data set description. $P$ is precipitation, $E$ is evapotranspiration, and $Q$ is runoff.

| Name | Variable(s) | Spatial Resolution | Temporal Resolution | Record Length | Data Type | Reference |
|---|---|---|---|---|---|---|
| CHMI | $P$ | Point | Daily | 1961-2020 | Stations | http://portal.chmi.cz |
| CRU TS v4.06 | $P$ | 1° | Monthly | 1901-2020 | Gauge-based | Harris et al. [2020] |
| E-OBS | $P$ | 0.125° | Daily | 1950-2020 | Gauge-based | Cornes et al. [2018] |
| ERA5-Land | $P, E, Q$ | 0.1° | Monthly | 1950-2020 | Reanalysis | Muñoz-Sabater et al. [2021] |
| GRDC | $Q$ | Point | Daily | 1921-2017 | Stations | www.bafg.de/GRDC |
| mHM | $E, Q$ | 0.125° | Daily | 1950-2020 | Model | Samaniego et al. [2010] |
| NCEP/NCAR R1 | $P, E, Q$ | T62 | Monthly | 1948-2020 | Reanalysis | Kalnay et al. [1996] |
| PREC/L | $P$ | 0.5° | Monthly | 1948-2020 | Gauge-based | Chen et al. [2002] |
| TerraClimate | $P, E, Q$ | 4 km | Monthly | 1958-2020 | Model | Abatzoglou et al. [2018] |

Line 173: Are there any supporting references to this similar approach? If yes, please provide the citations.

To the best of our knowledge there are no references for a similar approach. This is our proposed equation.

Line 180: It is okay to use the medians for excluding the outliers, but can you provide a supported plot to show the distribution of values as supplementary material?

Instead of adding a supplementary figure, Figure 4 will be revised from a histogram to a box plot. In the revised Figure 5 it can be seen that outliers are present only in the latter period (i.e., 1991-2020).

[Figure]

**Figure 5.** Box plots of spatial weighted average annual water fluxes over Czechia, where *P* is precipitation, *E* is evapotranspiration, *Q* is runoff, and *P* − *E* is precipitation minus evapotranspiration. Data are divided into two 30-year periods: 1961-1990 (blue) and 1991-2020 (yellow). Note that outliers are present only in the latter period (i.e., 1991-2020) as expected from the recorded severe drought of 2003.

Line 196-197: The demonstration is on the edge, as it is not all time is overestimated and under-estimated, only in some certain period.

It is true that overestimation or underestimation are not present at every single time step. The statements referred to the overall discrepancies as quantified by the 1981-2020 average. The text will be updated to reflect the revised evaluation values (see second major comment from Reviewer 1), also for clarity the text will be rephrased from: "mHM has the highest correlation for runoff, with R-squared circa 0.86 (Figure 2c), falling to the second highest for evapotranspiration (R-squared 0.7; Figure 2c). Interestingly, the values for the 30-year average in mHM underestimates runoff (Figure 2c) but overestimates evapotranspiration (Figure 2b)."

To: "mHM has the highest correlation for runoff, with R-squared circa 0.93 (Figure 2c)."

Figure 4: Can you use the line plot to show the trend as this is a time series for changes in hydrological variables, while a histogram may not be very straightforward?

The intent of the figure is to show the change between two climate normals, not the overall trend. Thus, the figure will revised into a box plot not a line pot (Figure 5 above on reply to comment "Line 180").

Figure 7-9: When you discuss the spatial distributions in different parts of Czechia maybe just focus on the one figure which is most representative as I see the spatial patterns are similar across Figure 7-9 and moved the rest figures to supplementary materials.

Figure 7 (now Figure 8) will be kept in the main manuscript and Figures 8 and 9 will be in the supplementary as Figure S3 and S4, respectively.

**Minor comments:**

Nine datasets? But in Table 1 there are ten datasets, right?

Nine data sets indeed. Table 1 will be revised and data type will be added (see major comment on Table 1)

Figure 5: Is it possible to zoom in on the y-axis limit because the boxes in the second and third rows are not clear?

The y-axis was modified as suggested:

[Figure]

**Figure 6.** Box plot of spatial weighted average monthly water fluxes over Czechia, where *P* is precipitation, *E* is evapotranspiration, *Q* is runoff, and *P* − *E* is precipitation minus evapotranspiration. Data are divided into two 30-year periods: 1961-1990 (blue) and 1991-2020 (yellow). Left column: TerraClimate (*P*), TerraClimate (*E*), and TerraClimate (*Q*). Middle column: mHM(E-OBS) (*P*), mHM (*E*), and mHM (*Q*). Right column: ERA5-Land (*P*), ERA5-Land (*E*), and ERA5-Land (*Q*).

**References**

Abatzoglou JT, Dobrowski SZ, Parks SA, Hegewisch KC (2018) TerraClimate, a high-resolution global dataset of monthly climate and climatic water balance from 1958–2015. Scientific Data 5(1):170191, DOI 10.1038/sdata.2017.191, number: 1 Publisher: Nature Publishing Group

Adler RF, Huffman GJ, Chang A, Ferraro R, Xie PP, Janowiak J, Rudolf B, Schneider U, Curtis S, Bolvin D (2003) The version-2 global precipitation climatology project (GPCP) monthly precipitation analysis (1979–present). Journal of hydrometeorology 4(6):1147–1167

Bosilovich MG, Robertson FR, Chen J (2011) Global energy and water budgets in MERRA. Journal of Climate 24(22):5721–5739

Bytheway JL, Kummerow CD (2013) Inferring the uncertainty of satellite precipitation estimates in data-sparse regions over land. Journal of Geophysical Research: Atmospheres 118(17):9524–9533, publisher: Wiley Online Library

Chen M, Xie P, Janowiak JE, Arkin PA (2002) Global land precipitation: A 50-yr monthly analysis based on gauge observations. Journal of Hydrometeorology 3(3):249–266

Cornes RC, van der Schrier G, van den Besselaar EJM, Jones PD (2018) An Ensemble Version of the E-OBS Temperature and Precipitation Data Sets. Journal of Geophysical Research: Atmospheres 123(17):9391–9409, DOI 10.1029/2017JD028200, URL https://onlinelibrary.wiley.com/doi/abs/10.1029/2017JD028200, _eprint: https://onlinelibrary.wiley.com/doi/pdf/10.1029/2017JD028200

Funk CC, Peterson PJ, Landsfeld MF, Pedreros DH, Verdin JP, Rowland JD, Romero BE, Husak GJ, Michaelsen JC, Verdin AP (2014) A quasi-global precipitation time series for drought monitoring. US Geological Survey Data Series 832(4):1–12

Gonzalez Miralles D, Holmes T, De Jeu R, Gash J, Meesters A, Dolman A (2011) Global land-surface evaporation estimated from satellite-based observations. Hydrology and Earth System Sciences pp 453–469

Harris I, Osborn TJ, Jones P, Lister D (2020) Version 4 of the CRU TS monthly high-resolution gridded multivariate climate dataset. Scientific data 7(1):1–18, publisher: Nature Publishing Group

Held IM, Soden BJ (2006) Robust Responses of the Hydrological Cycle to Global Warming. Journal of Climate 19(21):5686–5699, DOI 10.1175/JCLI3990.1, publisher: American Meteorological Society Section: Journal of Climate

Hong Y, Hsu KL, Sorooshian S, Gao X (2004) Precipitation estimation from remotely sensed imagery using an artificial neural network cloud classification system. Journal of Applied Meteorology 43(12):1834–1853

Huffman GJ, Bolvin DT, Nelkin EJ, Wolff DB, Adler RF, Gu G, Hong Y, Bowman KP, Stocker EF (2007) The TRMM multisatellite precipitation analysis (TMPA): Quasi-global, multiyear, combined-sensor precipitation estimates at fine scales. Journal of hydrometeorology 8(1):38–55

Huffman GJ, Adler RF, Bolvin DT, Gu G (2009) Improving the global precipitation record: GPCP version 2.1. Geophysical Research Letters 36(17)

Huffman GJ, Adler RF, Bolvin DT, Nelkin EJ (2010) The TRMM multi-satellite precipitation analysis (TMPA). In: Satellite rainfall applications for surface hydrology, Springer, pp 3–22

Joyce RJ, Janowiak JE, Arkin PA, Xie P (2004) CMORPH: A method that produces global precipitation estimates from passive microwave and infrared data at high spatial and temporal resolution. Journal of hydrometeorology 5(3):487–503

Jung M, Reichstein M, Ciais P, Seneviratne SI, Sheffield J, Goulden ML, Bonan G, Cescatti A, Chen J, de Jeu R, Dolman AJ, Eugster W, Gerten D, Gianelle D, Gobron N, Heinke J, Kimball J, Law BE, Montagnani L, Mu Q, Mueller B, Oleson K, Papale D, Richardson AD, Roupsard O, Running S, Tomelleri E, Viovy N, Weber U, Williams C, Wood E, Zaehle S, Zhang K (2010) Recent decline in the global land evapotranspiration trend due to

limited moisture supply. Nature 467(7318):951–954, DOI 10.1038/nature09396, number: 7318 Publisher: Nature Publishing Group

Kalnay E, Kanamitsu M, Kistler R, Collins W, Deaven D, Gandin L, Iredell M, Saha S, White G, Woollen J (1996) The NCEP/NCAR 40-year reanalysis project. Bulletin of the American meteorological Society 77(3):437–472

Landerer FW, Swenson S (2012) Accuracy of scaled GRACE terrestrial water storage estimates. Water resources research 48(4), publisher: Wiley Online Library

Mitchell TD, Jones PD (2005) An improved method of constructing a database of monthly climate observations and associated high-resolution grids. International Journal of Climatology: A Journal of the Royal Meteorological Society 25(6):693–712, publisher: Wiley Online Library

Mu Q, Heinsch FA, Zhao M, Running SW (2007) Development of a global evapotranspiration algorithm based on MODIS and global meteorology data. Remote sensing of Environment 111(4):519–536, publisher: Elsevier

Munier S, Aires F (2018) A new global method of satellite dataset merging and quality characterization constrained by the terrestrial water budget. Remote Sensing of Environment 205:119–130, publisher: Elsevier

Muñoz-Sabater J, Dutra E, Agustí-Panareda A, Albergel C, Arduini G, Balsamo G, Boussetta S, Choulga M, Harrigan S, Hersbach H, Martens B, Miralles DG, Piles M, Rodríguez-Fernández NJ, Zsoter E, Buontempo C, Thépaut JN (2021) ERA5-Land: a state-of-the-art global reanalysis dataset for land applications. Earth System Science Data 13(9):4349–4383, DOI 10.5194/essd-13-4349-2021, publisher: Copernicus GmbH

Pan M, Sahoo AK, Troy TJ, Vinukollu RK, Sheffield J, Wood EF (2012) Multisource estimation of long-term terrestrial water budget for major global river basins. Journal of Climate 25(9):3191–3206

Reichle R (2012) The MERRA-land data product (version 1.2). GMAO Off Note 3

Rienecker MM, Suarez MJ, Gelaro R, Todling R, Bacmeister J, Liu E, Bosilovich MG, Schubert SD, Takacs L, Kim GK (2011) MERRA: NASA's modern-era retrospective analysis for research and applications. Journal of climate 24(14):3624–3648

Rodell M, Beaudoing HK, L'Ecuyer T, Olson WS, Famiglietti JS, Houser PR, Adler R, Bosilovich MG, Clayson CA, Chambers D, others (2015) The observed state of the water cycle in the early twenty-first century. Journal of Climate 28(21):8289–8318

Roderick M, Sun F, Lim WH, Farquhar G (2014) A general framework for understanding the response of the water cycle to global warming over land and ocean. Hydrology and Earth System Sciences 18(5):1575–1589, publisher: Copernicus GmbH

Sahoo AK, Pan M, Troy TJ, Vinukollu RK, Sheffield J, Wood EF (2011) Reconciling the global terrestrial water budget using satellite remote sensing. Remote Sensing of Environment 115(8):1850–1865, publisher: Elsevier

Samaniego L, Kumar R, Attinger S (2010) Multiscale parameter regionalization of a grid-based hydrologic model at the mesoscale. Water Resources Research 46(5), DOI 10.1029/2008WR007327, _eprint: https://onlinelibrary.wiley.com/doi/pdf/10.1029/2008WR007327

Schneider U, Becker A, Finger P, Meyer-Christoffer A, Rudolf B, Ziese M (2011) GPCC full data reanalysis version 6.0 at 0.5: monthly land-surface precipitation from rain-gauges built on GTS-based and historic data. GPCC Data Rep, doi 10

Schneider U, Becker A, Finger P, Meyer-Christoffer A, Ziese M, Rudolf B (2014) GPCC's new land surface precipitation climatology based on quality-controlled in situ data and its role in quantifying the global water cycle. Theoretical and Applied Climatology 115(1-2):15–40, publisher: Springer

Sheffield J, Wood EF (2007) Characteristics of global and regional drought, 1950–2000: Analysis of soil moisture data from off-line simulation of the terrestrial hydrologic cycle. Journal of Geophysical Research: Atmospheres 112(D17), publisher: Wiley Online Library

Sheffield J, Goteti G, Wood EF (2006) Development of a 50-year high-resolution global dataset of meteorological forcings for land surface modeling. Journal of climate

19(13):3088–3111

Sheffield J, Wood EF, Munoz-Arriola F (2010) Long-Term Regional Estimates of Evapotranspiration for Mexico Based on Downscaled ISCCP Data. Journal of Hydrometeorology 11(2):253–275, DOI 10.1175/2009JHM1176.1, publisher: American Meteorological Society Section: Journal of Hydrometeorology

Simmons A (2006) ERA-Interim: New ECMWF reanalysis products from 1989 onwards. ECMWF newsletter 110:25–36

Swenson S, Wahr J (2006) Estimating Large-Scale Precipitation Minus Evapotranspiration from GRACE Satellite Gravity Measurements. Journal of Hydrometeorology 7(2):252–270, DOI 10.1175/JHM478.1, publisher: American Meteorological Society Section: Journal of Hydrometeorology

Trenberth KE (2011) Changes in precipitation with climate change. Climate Research 47(1-2):123–138

Turk JT, Mostovoy GV, Anantharaj V (2010) The NRL-blend high resolution precipitation product and its application to land surface hydrology. In: Satellite Rainfall Applications for Surface Hydrology, Springer, pp 85–104

Vinukollu RK, Wood EF, Ferguson CR, Fisher JB (2011) Global estimates of evapotranspiration for climate studies using multi-sensor remote sensing data: Evaluation of three process-based approaches. Remote Sensing of Environment 115(3):801–823

Willmott CJ, Matsuura K (2001) Terrestrial Air Temperature and Precipitation: Monthly and Annual Time Series (1950 - 1999). University of Delaware

Zhang K, Kimball JS, Nemani RR, Running SW (2010) A continuous satellite-derived global record of land surface evapotranspiration from 1983 to 2006. Water Resources Research 46(9), publisher: Wiley Online Library

Zhang Y, Pan M, Wood EF (2016) On creating global gridded terrestrial water budget estimates from satellite remote sensing. In: Remote Sensing and Water Resources, Springer, pp 59–78

---

## Author Comment (AC3)

**Reply to Reviewer 3**

This paper presents analyses of the water budget and water cycle for Czechia. Overall I found that the work is interesting and well written. My major comment is around the definition of the score used for the ranking of the different data set combinations and how this was derived and justified. For example the score only accounts for the anomalies and the correlations but does not consider bias in the products. This is very evident from Figure 4 where ERA5-land has substantially higher estimates of both P and ET and therefore its anomalies are similar to the other products. But presumably in some applications consistent biases may be problematic even if the anomalies are ok (e.g. water allocations or environmental flows). I think that the authors need to do far more to consider the sensitivity of the dataset ranking to the definition of the score.

We thank the reviewer for their constructive and encouraging comments. As correctly pointed out the score does not account for any biases in the products. However, if precipitation and evapotranspiration are over- or underestimated simultaneously then the overall water budget closure is not significantly affected. The metric proposed herein aims to rank multi-source data combinations to determine how well a given combination of data sets closes the water budget. It is a method that can be used to easily and quickly filter out the data set combinations providing implausible results and then be complemented with additional analyses that consider the bias as we did in the original manuscript. We agree that the approach introduced in our work might not be the best suited for different applications that need to quantify absolute values rather than anomalies in water fluxes. The main aim of our work is not to benchmark the different data sets analyzed herein but to demonstrate how different can become the water cycles depicted by each of them. To clarify this, we will add in the revised manuscript the following:

"Our evaluation of individual water cycle components is cohesive with previous literature. Although the data products assessed herein have been previously analyzed at multiple spatial scales, this is done under a univariate perspective, that does not consider the ability of the data sets to reproduce the water cycle and its changes as a whole in a structurally plausible manner. This is easily denoted by the fact that even though mHM's performance was the best for all water cycle components evaluated using high-quality observational references, the best data set combination ranking is actually TerraClimate exclusive (i.e., all flux estimates from the same data set). Note that the score metric and ranking framework proposed herein serve as a method that can easily and quickly filter out the data set combinations providing implausible results. It should be remarked that this ranking framework acts as an initial assessment to be complemented with additional analyses because the score metric does not account for any biases in the products. Expressly because our work aims not to benchmark the different data sets analyzed herein but to demonstrate how different can become the water cycles depicted by each of them."

**Minor comments:**

Figure 1: shading is difficult to interpret and I think it would be easier to use hatching or just label the rivers

Figure 1 will be revised as suggested:

[Figure]

**Figure 1.** The three drainage basins within Czechia's boundaries. Elbe (light gray shade), Danube (striped dark gray shade), and Oder (no shade).

Line 163: Would be interesting to do the analyses for the three main drainage basins.

The following figure with the corresponding text will be added:

"The water cycle budget is meant to close over hydrological units. Accordingly, we examined the water fluxes of the data sets with the best evaluation over the subbasins enclosed by the Czech administrative borders (Figure 3). For simplicity, we will refer to them as the Danube basin inside Czechia, the Elbe basin inside Czechia, and the Oder basin inside Czechia. It can be seen that within each data set, no extremely deviant behavior is exhibited between basins or at the country level. In other words, the precipitation time series depicted by TerraClimate for Czechia is similar to the one depicted for the Danube, Elbe, and Oder basins inside Czechia. Comparing data sets, however, it is evident that ERA5-Land is different. At first glance, we evince higher magnitudes for ERA5-Land precipitation and evapotranspiration, yet the residuals do not appear to be that far off from those of mHM or TerraClimate. It is not until we look at the cumulative sum of the residuals that we can distinguish ERA5-Land water budget residuals are nonstationary with a decreasing trend."

[Figure]

**Figure 3.** Spatial weighted average annual water fluxes over Czechia (first row), Danube basin inside Czechia (second row), Elbe basin inside Czechia (third row), and Oder basin inside Czechia (fourth row). Where $P$ is precipitation in blue, $E$ is evapotranspiration in green, $Q$ is runoff in purple, $\xi$ is the residual $(P - E - Q)$ in black, and cumsum($\xi$) is the cumulative sum of the residual in orange. Left column: TerraClimate ($P$), TerraClimate ($E$), and TerraClimate ($Q$). Middle column: mHM(E-OBS) ($P$), mHM ($E$), and mHM ($Q$). Right column: ERA5-Land ($P$), ERA5-Land ($E$), and ERA5-Land ($Q$).

Line 170: would be good to explicitly note that you are doing the closure each year here and then Ri is the average of Rj for j in 1:60

To explicitly describe the average residual we will modify the manuscript from: "A success metric widely used among several studies is getting the budget closure residual ($R$) as close to zero as possible. Herein, we define the budget closure residual as follows:

$$R = P - E - Q \qquad (1)$$

where $P$ is precipitation, $E$ is evapotranspiration, and $Q$ is runoff. Thus, we have 96 distributions of 60 annual values each. The ranking of a given data set combination was determined via:

$$Ranking = \frac{|\overline{R_i}|\sigma_{R_i}}{(cor(P_i - E_i, Q_i)cor(P_i, P_o)cor(E_i, E_o)cor(Q_i, Q_o))^2} \qquad (2)$$

where $|\overline{R_i}|$ is the absolute value of the mean of the 60 annual residuals for the $i$-th combination, $\sigma_{R_i}$ is the standard deviation of the 60 annual residuals for the $i$-th combination, $cor(P_i - E_i, Q_i)$

is the correlation between $P-E$ and $Q$ for the $i$-th combination, $cor(P_i, P_o)$ is the correlation between $P$ of the $i$-th combination and the precipitation evaluation reference, $cor(E_i, E_o)$ is the correlation between $E$ of the $i$-th combination and the evapotranspiration evaluation reference, and $cor(Q_i, Q_o)$ is the correlation between $Q$ of the $i$-th combination and the runoff evaluation reference."

To: "A success metric widely used among several studies is getting the budget closure residual ($\xi$) as close to zero as possible. Herein, we define the budget closure residual as follows:

$$\xi_n = P_n - E_n - Q_n \tag{3}$$

where $P_n$ is precipitation, $E_n$ is evapotranspiration, and $Q_n$ is runoff for a given year $n$. Thus, we have 60 annual values for each of the 96 possible combinations. Under steady state conditions the mean of these residuals should tend to zero:

$$\overline{\xi_i} = \frac{\sum_{n=1}^{N} \xi_n}{N} \to 0 \tag{4}$$

where $\overline{\xi_i}$ is the mean of the $N = 60$ annual residuals for the $i$-th combination. The score to be used in the ranking of a given data set combination was determined via:

$$score = \frac{|\overline{\xi_i}|\sigma_{\xi_i}}{(cor(P_i - E_i, Q_i)cor(P_i, P_o)cor(E_i, E_o)cor(Q_i, Q_o))^2} \tag{5}$$

where $|\overline{\xi_i}|$ is the absolute value of the mean of the 60 annual residuals for the $i$-th combination, $\sigma_{\xi_i}$ is the standard deviation of the 60 annual residuals for the $i$-th combination, $cor(P_i-E_i, Q_i)$ is the correlation between $P-E$ and $Q$ for the $i$-th combination, $cor(P_i, P_o)$ is the correlation between $P$ of the $i$-th combination and the precipitation evaluation reference, $cor(E_i, E_o)$ is the correlation between $E$ of the $i$-th combination and the evapotranspiration evaluation reference, and $cor(Q_i, Q_o)$ is the correlation between $Q$ of the $i$-th combination and the runoff evaluation reference."

Equation 2: this isn't actually the ranking but a score that is then used for ranking so I think all the text associated with the equation needs to be updated.

The text will be rephrased from: "The ranking of a given data set combination was determined via:"

To: "The score to be used in the ranking of a given data set combination was determined via:"

Figure 3 - we can't see most of the distributions. I don't think this is a useful presentation of the data. What are the units for the budget residual?

Figure 3 (now Figure 4) will be modified to include only the distributions listed in table 2. The original figure with all the distributions will be placed in the supplementary material as Figure S2.

[Figure]

**Figure 4.** Empirical distribution of the data set combinations listed on 2 colored based on their ranking as determined by Equation 5. The color gradient goes from higher ranked combinations colored in shades green to lower ranked combinations colored in shades of brown.

Figure 5 - wrong colours mentioned in caption. I am surprised by the results shown in figure 5 as there is less difference between the different models than implied by Figure 4 where ERA5 is substantially wetter and higher ET. I think you could dig further into this.

We thank the reviewer for their detailed attention and corresponding suggestions. Captions will be revised to describe the appropriate colors. The story regarding water cycle changes depends on the data set of choice and the time scale. These differences tend to be overlooked when annual averages are being compared, but when it comes to annual totals, the small discrepancies add up, leading to such results. We further highlight some substantial inconsistency in the ERA5-Land data (Figure 3). It appears that the cumulative sum of the water budget residual in ERA5-Land declines monotonically in time, implying some systematic bias in the water budget closure. Even though that approximately 500 mm over 60 years might be considered a relatively small amount, it raises further questions about the applicability of ERA5-Land in hydrological studies and therefore, extra caution should be taken when the widely-used reanalysis data product is employed.

---

## Author Response (AR2)

**Reply to Reviewers**

Thank you for sending your revised manuscript, which has been read by myself and by two of the same original reviewers. Both reviewers agree the significance and presentation of the manuscript are good and that it should be accepted with some small revisions. This aligns with my own reading of the work, and I would like to invite you to kindly make these revisions. Additionally, please could you add a header at the top of your Supplementary materials with the manuscript details.

We thank the reviewers and the editor for providing further constructive comments and suggestions. We have revised the manuscript accordingly and added a header at the top of the Supplementary material. In the following, we provide detailed replies to all comments and discuss changes to the main manuscript.

**Reviewer 2**

We thank the reviewer for their detailed comments.

**Minor revision**

1. Figure 1 caption, what is the meaning of clear for Oder, the red borders?

Figure 1 and its caption were revised to improve its clarity.

**Figure 1.** The three drainage basins within Czechia's administrative boundaries (red line). Elbe (light gray shade), Danube (black stripes), and Oder (dark gray points).

2. Line 90-103: The authors introduce CHMI and GRDC as assessment references, can you briefly describe in a sentence or two why these two are more appropriate as references than other datasets in section 2.2.1? Have previous studies been compared or evaluated between these datasets presented in this paper?

To further support the selection of CHMI and GRDC as appropriate references rather than any of the remaining data sets included in our study, the following text was added at the beginning of Section 2.2.1:

"As evaluation references, we relied solely on ground station data sets. A distinct advantage of station data over hydrological models or reanalyses is their capability to capture detailed and localized information. These in-situ measurements directly reflect the local climatic conditions, offering a more accurate representation of the water cycle."

To the best of our knowledge there are no previous studies comparing all these data sets simultaneously nor in the context of our study.

3. Figure 2: it might be clearer to zoom in on the y-axis in Figure 2b.

The y-axis was modified as suggested: